# On the Convergence of Single-Timescale Actor-Critic

**Navdeep Kumar**[*]
Technion

**Priyank Agrawal,**
Columbia University

**Giorgia Ramponi**
University of Zurich

**Kfir Levy**
Technion

**Shie Mannor**
Technion

## Abstract

We analyze the global convergence of the single-timescale actor-critic (AC) algorithm for the infinite-horizon discounted Markov Decision Processes (MDPs) with finite state spaces. To this end, we introduce an elegant analytical framework for handling complex, coupled recursions inherent in the algorithm. Leveraging this framework, we establish that the algorithm converges to an $\epsilon$-close **globally optimal** policy with a sample complexity of $O(\epsilon^{-3})$. This significantly improves upon the existing complexity of $O(\epsilon^{-2})$ to achieve $\epsilon$-close **stationary policy**, which is equivalent to the complexity of $O(\epsilon^{-4})$ to achieve $\epsilon$-close **globally optimal** policy using gradient domination lemma. Furthermore, we demonstrate that to achieve this improvement, the step sizes for both the actor and critic must decay as $O(k^{-\frac{2}{3}})$ with iteration $k$, diverging from the conventional $O(k^{-\frac{1}{2}})$ rates commonly used in (non)convex optimization.

## 1 Introduction

Actor-critic algorithm, initially introduced in Konda and Tsitsiklis (1999), consist of two key components: the actor, which refines the policy towards an optimal solution based on feedback from the critic, and the critic, which evaluates the value of the current policy (specifically the Q-value). It has been adapted in various forms Schulman et al. (2017) and have emerged as one of the most successful methods in reinforcement learning (Mnih et al., 2015; Silver et al., 2017; OpenAI et al., 2019; Schrittwieser et al., 2020).

Despite their remarkable empirical success, the theoretical convergence of actor-critic algorithms is not well understood. One line of research explores a two-time-scale version in which the actor and the critic are effectively decoupled, greatly simplifying the analyses. This can be achieved via a double-loop version, where the critic evaluates the policy in the inner loop, and the actor updates the policy in the outer loop (Yang et al., 2019; Agarwal et al., 2020; Wang et al., 2022; Kumar et al., 2023; Wang et al., 2019), or via a single-loop structure, but the critic updates much faster than the actor (Borkar, 2022). In the later setup, the ratio of the learning rates of the actor and critic tends to zero with the number of iterations. Essentially, the critic perceives the actor as nearly stationary, while the actor views the critic as almost converged. Konda and Tsitsiklis (1999); Bhatnagar et al. (2009); Chen et al. (2023); Hong et al. (2022); Wu et al. (2022); Xu et al. (2020b). It is important to note that both frameworks are artificial constructs to ease the analysis, but they are often sample-inefficient and therefore seldom used in practical implementations (Olshevsky and Gharesifard, 2023).

In this work, we focus on a single time-scale actor-critic framework where both the actor and the critic are updated with each sample using similar step sizes Sutton and Barto (2018). While this framework

---

[*]corresponding author email `navdeep.kumar@zohomail.in`

39th Conference on Neural Information Processing Systems (NeurIPS 2025).

is more versatile and practical, but the theoretical analysis of single-time actor-critic algorithms faces significant challenges due to the strong coupling between the actor and critic. Since both components evolve inseparably together with similar rates, the analytical challenge lies in understanding a stable error propagation schedule.

For the first time, Castro and Meir (2009) established asymptotic convergence of the single time scale actor critic to a neighborhood of an optimal value. This was followed by the recent works Chen et al. (2021); Olshevsky and Gharesifard (2023); Chen and Zhao (2024) demonstrating a sample complexity of $O(\epsilon^{-2})$ for achieving an $\epsilon$-close **stationary** policy, where the squared norm of the gradient of the return is less than $\epsilon$, under various settings. This corresponds to a sample complexity of $O(\epsilon^{-4})$ for achieving an $\epsilon$-close globally **optimal** policy (see Proposition 3.2). The question of whether this $O(\epsilon^{-4})$ complexity can be further improved remains open, and this paper provides a favorable answer.

In this work, we first formulate the recursions for actor and critic errors which are quite complex. None of the actor and critic errors are monotonically decreasing. We then identify a Lyaponov term (sum of actor error and squared of critic error), and obtain its recursions independent of all the other terms. This Lyapunov recursion is monotonically decreasing but more challenging than in the exact gradient case found in Xiao (2022); Zhang et al. (2020), due to the presence of a time-dependent learning rate. To address this, we develop an elegant ODE domination methodology for solving these recursions, yielding significantly improved bounds.

**Our contributions are summarized as follows:**

1. **Improved Global Convergence Rate:** We establish a sharper global convergence result for single-timescale actor-critic algorithms in softmax-parameterized discounted MDPs. Our analysis shows a sample complexity of $O(\epsilon^{-3})$ to compute an $\epsilon$-optimal policy, improving upon the prior best rate of $O(\epsilon^{-4})$.

2. **ODE-Based Methodology with Direct Global Guarantees:** Our core technical innovation is a streamlined ODE-based analysis for resolving the interdependent actor and critic updates. Unlike previous approaches that first bound convergence to stationary points (e.g., $O(\epsilon^{-2})$ for $\epsilon$-stationary policies), we directly bound the global sub-optimality gap $J^* - J^{\pi_k}$.

3. **Broad Applicability of Techniques:** The techniques developed are concise and modular, and may extend naturally to related settings such as minimax optimization, bi-level optimization, robust MDPs, and multi-agent reinforcement learning and could be of independent interest.

## 1.1 Related works

Policy gradient based methods Sutton and Barto (2018); Schulman et al. (2015); Mnih et al. (2015) have been well used in practice with empirical success exceeding beyond the value-based algorithms Auer et al. (2008); Azar et al. (2017); Jin et al. (2018); Agrawal and Agrawal (2024); Agrawal et al. (2025). Naturally, its convergence properties of policy gradient has been of a great interests. Only, asymptotic convergence of policy gradient has been well-established in Williams (1992); Sutton et al. (1999); Kakade (2001); Baxter and Bartlett (2001) until very recently as summarized below.

**Projected Policy Gradient (PPG):** Given oracle access to gradient, Bhandari and Russo (2024); Agarwal et al. (2020) established global convergence of the projected policy gradient (tabular setting) with an iteration complexity of $O(\epsilon^{-2})$ in discounted reward setting. Following up, an improved recursion analysis, led to complexity of $O(\epsilon^{-1})$ Xiao (2022). Recently, Liu et al. (2024a) obtained an linear convergence was obtained for an large enough learning rate and also for aggressively increasing step sizes. Further, PPG is proven to find global optimal policy in finite steps Liu et al. (2024b).

**Softmax Parametrized Policy Gradient** Often in practice, parametrized policies are used and softmax is an one of the most popular parametrization. Softmax policy gradient (1) enjoys iteration complexity of $O(\epsilon^{-1})$ for global convergence Mei et al. (2022); Liu et al. (2024a). This complexity is matching with lower bound of $O(\epsilon^{-1})$ established in Mei et al. (2022); Liu et al. (2024a).

**Stochastic Policy Gradient Descent** Often the gradient is not available in practice, and is estimated via samples. Vanilla SGD (stochastic gradient descent) and stochastic variance reduced

gradient descent (SVRGD) has sample complexity of $O(\epsilon^{-2})$ and $O(\epsilon^{-\frac{5}{3}})$ respectively, for achieving $\|\nabla J^\pi\|_2^2 \leq \epsilon$ (where $J^\pi$ is return of the policy $\pi$) Xu et al. (2020a). This local convergence yields global convergence of iteration complexity of $O(\epsilon^{-4})$, $O(\epsilon^{-\frac{10}{3}})$ for SGD and SVRGD respectively using Proposition 3.2. Further, SGD achieves second order stationary point with an iteration complexity of $O(\epsilon^{-9})$ Zhang et al. (2020).

**Single Time Scale Actor-critic Algorithm:**   It is a class of algorithms where critic (gradient, value function) and actor (policy) is updated simultaneously. This is arguably the most popular algorithms used in many variants in practice Konda and Tsitsiklis (1999); Bhatnagar et al. (2009); Schulman et al. (2015, 2017). Castro and Meir (2009) first established asymptotic convergence of the single time scale actor-critic algorithm. Later, Olshevsky and Gharesifard (2023); Chen and Zhao (2024); Olshevsky and Gharesifard (2023) established the local convergence of single time-scale actor-critic algorithm with ( see Table 1) sample complexity of $O(\epsilon^{-2})$ for achieving $\|\nabla J^\pi\|_2^2 \leq \epsilon$. This yields global convergence ($J^* - J^\pi \leq \epsilon$, where $J^*$ optimal return) with sample complexity of $O(\epsilon^{-4})$ using Gradient Domination Lemma as shown in Proposition 3.2 Olshevsky and Gharesifard (2023).

The main limitation of the analysis in Chen and Zhao (2024) is that it treats the policy optimization objective as a generic smooth non-convex function and follows the standard approach of bounding the average squared gradient norm. This ignores the gradient domination structure, which, if applied only at the end, yields a weaker rate of $O(\epsilon^{-4})$. Our key innovation is to explicitly exploit this structure when constraining the iteration-wise drift of the actor. Doing so required developing new techniques to handle the resulting interdependent recursions, leading to stronger results. In summary, our analysis is more tailored to RL by effectively leveraging the gradient domination property, unlike the standard smooth optimization approach used in prior work.

**Two Time Scale (/Double Loop) Actor Critic Algorithm.**   First, Konda and Tsitsiklis (1999) showed convergence of actor-critic algorithm to a stationary point using two time scale analysis of Borkar (2022). The work Gaur et al. (2024) establishes $O(\epsilon^{-3})$ sample complexity of a actor-critic algorithm variant (see Algorithm 1 Gaur et al. (2024)). The algorithm uses $O(\epsilon^{-3})$ new samples for the global convergence. However, it maintains the buffer of $O(\epsilon^{-2})$ samples at each iteration. For achieving $\epsilon$-close global optimal policy, the algorithm requires $O(\epsilon^{-1})$ iteration, and each iteration repeatedly uses the samples from the buffer, $O(\epsilon^{-4})$ many times. In conclusion, the algorithm uses $O(\epsilon^{-3})$ new samples, using them $O(\epsilon^{-5})$ times in total, thereby significantly inflating the memory requirements and computational complexity. In comparison, our algorithm does not use any buffer and use new sample in each iteration.

**Natural Actor Critic (NAC) Algorithms.**   NAC algorithm is another class of algorithms Amari (1998); Kakade (2001); Bagnell and Schneider (2003); Peters and Schaal (2008); Bhatnagar et al. (2009) proposed to make the gradient updates independent of different policy parameterizations. It has linear convergence rate (iteration complexity of $O(\log \epsilon^{-1})$) under exact gradient setting Bhatnagar et al. (2009) which is much faster the vanilla gradient descent. Similarly, the sample based NAC algorithms Ganesh et al. (2024) also enjoys better sample complexity of $O(\epsilon^{-2})$. Xu et al. (2020b) establishes the global convergence of the natural actor-critic algorithm with a sample complexity of $O(\epsilon^{-4})$ in discounted reward MDPs. However, the natural actor-critic algorithm demands additional computations, which can be challenging. Yuan et al. (2022) too establishes global convergence with sample complexity of $O(\epsilon^{-3})$, however, it requires an additional structural assumption on the problem which is highly restrictive. However, NAC requires the inversion of the Fisher Information Matrix (FIM) in the update rule. This inverse computation makes the implementation difficult and sometimes unfeasible (for an instance, FIM is not invertible in direct parametrization, if $d^\pi(s) = 0$ for some $s$). We note that actor-critic is a very different algorithm than NAC, arguably the most useful and versatile, hence deserving its own independent study.

## 2   Preliminaries

We consider the class of infinite horizon discounted reward MDPs with finite state space $\mathcal{S}$ and finite action space $\mathcal{A}$ with discount factor $\gamma \in [0, 1)$ Sutton and Barto (2018); Puterman (1994). The underlying environment is modeled as a probability transition kernel denoted by $P \in (\Delta \mathcal{A})^{\mathcal{S} \times \mathcal{A}}$. We consider the class of randomized policies $\Pi = \{\pi : \mathcal{S} \to \Delta \mathcal{A}\}$, where a policy $\pi$ maps each state to a probability vector over the action space. The transition kernel corresponding to a policy $\pi$

Table 1: Related Work: Sample Complexity of Single Time Scale Actor Critic

| Work | Convergence | Sample Complexity | Actor Step size $\eta_k$ | Critic Step size $\beta_k$ | Sampling |
|---|---|---|---|---|---|
| Olshevsky and Gharesifard (2023),Chen et al. (2021) | $\|\nabla J^\pi\| \le \epsilon$ | $O(\epsilon^{-4})$ | $k^{-\frac{1}{2}}$ | $k^{-\frac{1}{2}}$ | i.i.d. |
| Chen and Zhao (2024) | $\|\nabla J^\pi\| \le \epsilon$ | $O(\epsilon^{-4})$ | $k^{-\frac{1}{2}}$ | $k^{-\frac{1}{2}}$ | Markovian |
| **Ours** | $J^* - J^\pi \le \epsilon$ | $O(\epsilon^{-3})$ | $k^{-\frac{2}{3}}$ | $k^{-\frac{2}{3}}$ | i.i.d. |

$\|\nabla J^\pi\| \le \epsilon \implies J^* - J^\pi \le c\epsilon$ for some constant $c$, see Proposition 3.2. These works are for different settings such average reward, discounted reward, finite state space, and infinite state space, please refer to the individual work for more details.

is represented by $P^\pi : \mathcal{S} \to \mathcal{S}$, where $P^\pi(s'|s) = \sum_{a \in \mathcal{A}} \pi(a|s)P(s'|s,a)$ denotes the single step probability of moving from state $s$ to $s'$ under policy $\pi$. Let $R(s,a) \in [-1,1]$ denote the single step reward obtained by taking action $a \in \mathcal{A}$ in state $s \in \mathcal{S}$. The single-step reward associated with a policy $\pi$ at state $s \in \mathcal{S}$ is defined as $R^\pi(s) = \sum_{a \in \mathcal{A}} \pi(a|s)R(s,a)$. The discounted average reward (or return) $J^\pi$ associated with a policy $\pi$ is defined as:

$$J^\pi = \mathbb{E}\left[\sum_{n=0}^\infty \gamma^n R^\pi(s_k) \mid \pi, P, s_0 \sim \mu\right] = \mu^T(I - \gamma P^\pi)^{-1}R^\pi,$$

where $\mu \in \Delta \mathcal{S}$ denotes the initial state distribution. It can be alternatively expressed as $J^\pi = (1-\gamma)^{-1}\sum_{s \in \mathcal{S}} d^\pi(s)R^\pi(s)$, where $d^\pi = (1-\gamma)\mu^T(I - \gamma P^\pi)^{-1}$ is the stationary measure under the transition kernel $P^\pi$. Value function $v^\pi := (I - \gamma P^\pi)^{-1}R^\pi$ satisfies the following Bellman equation $v^\pi = R^\pi + \gamma P^\pi v^\pi$ (Puterman, 1994; Bertsekas, 2007). The Q-value function $Q^\pi \in \mathbb{R}^{\mathcal{S} \times \mathcal{A}}$ associated with a policy $\pi$ is defined as $Q^\pi(s,a) = R(s,a) + \gamma \sum_{s' \in \mathcal{S}} P(s'|s,a)v^\pi(s')$ for all $(s,a) \in \mathcal{S} \times \mathcal{A}$. For simplicity, we will also assume $\|R\|_\infty \le 1$.

In this paper, we consider soft-max policy parameterized by $\theta \in \mathbb{R}^{\mathcal{S} \times \mathcal{A}}$ as $\pi_\theta(a|s) = \frac{e^{\theta(s,a)}}{\sum_a e^{\theta(s,a)}}$ Mei et al. (2022). The objective is to obtain an optimal policy $\pi^*$ that maximizes the return $J^\pi$. We denote $J^*$ as a shorthand for the optimal return $J^{\pi^*}$. The exact policy gradient update is given as

$$\theta_{k+1} := \theta_k + \eta_k \nabla J^{\pi_{\theta_k}}, \tag{1}$$

where $\eta_k$ is the learning rate, in most vanilla form Sutton and Barto (2018). The policy gradient can be derived as

$$\frac{\partial J^{\pi_\theta}}{\partial \theta(s,a)} = (1-\gamma)^{-1}d^{\pi_\theta}(s)\pi_\theta(a|s)A^{\pi_\theta}(s,a),$$

where $A^\pi(s,a) := Q^\pi(s,a) - v^\pi(s)$ is advantage function Mei et al. (2022). The return $J^{\pi_\theta}$ is a highly non-concave function, making global convergence guarantees for the above policy gradient method very challenging. However, the return $J^{\pi_\theta}$ is $L = \frac{8}{(1-\gamma)^3}$-smooth with respect to $\theta$ Mei et al. (2022).

**Lemma 2.1.** *(Gradient Domination Lemma, Mei et al. (2022)) The sub-optimality is upper bounded by the norm of the gradient as*

$$\|\nabla J^{\pi_{\theta_k}}\|_2 \ge \frac{c}{\sqrt{S}C_{PL}}\left[J^* - J^{\pi_{\theta_k}}\right],$$

*where $C_{PL} = \max_k \|\frac{d^{\pi^*}}{d^{\pi_{\theta_k}}}\|_\infty$ is mismatch coefficient and $c = \min_k \min_s \pi_{\theta_k}(a^*(s)|s)$,*

The result states that the norm of the gradient vanishes only when the sub-optimality is zero. In other words, the gradient is zero only at the optimal policies. This, combined with the Sufficient Increase Lemma, directly leads to the global convergence of the policy gradient update rule in (1).

However, the above lemma requires the mismatch coefficient $C_{PL}$ to be bounded, which can be ensured by setting the initial distribution $\mu(s) > 0$ for all states. Unfortunately, failure to ensure $\mu \succ 0$ may lead to local solutions Kumar et al. (2024). Additionally, the result requires the constant $c$ to be strictly greater than zero. This condition can be satisfied by initializing the parameterization with $\theta_0 = 0$ or by ensuring it remains bounded. Furthermore, as the iterates progress towards an optimal policy, the constant $c$ remains bounded away from zero.

## 3  Main

In this work, we focus on the convergence of the widely used single time-scale actor-critic algorithm (1), where the actor (policy) and critic (value function) are updated simultaneously Konda and Tsitsiklis (1999); Sutton and Barto (2018); Chen et al. (2021); Olshevsky and Gharesifard (2023); Chen and Zhao (2024). Notably, this algorithm operates with a single sample per iteration, without relying on batch processing or maintaining an experience replay buffer.

---

**Algorithm 1** Single Time Scale Actor Critic Algorithm

---

**Input**: Stepsizes $\eta_k, \beta_k$

1: **while** not converged; $k = k + 1$ **do**
2:  Sample $s \sim d^{\pi_{\theta_k}}, a \sim \pi_{\theta_k}(\cdot|s)$ and get the next state-action $s' \sim P(\cdot|s, a), a' \sim \pi_{\theta_k}(\cdot|s')$ .

3:  Policy update:
$$\theta_{k+1}(s, a) = \theta_k(s, a) + \eta_k (1 - \gamma)^{-1} A(s, a),$$
   where $A(s, a) = Q(s, a) - v(s)$ and $v(s) = \sum_a \pi_{\theta_k}(a|s) Q(s, a)$.

4:  Q-value update:
$$Q(s, a) = Q(s, a) + \beta_k \left[ R(s, a) + \gamma Q(s', a') - Q(s, a) \right].$$

5: **end while**

---

Our objective is to derive a policy $\pi$ that maximizes the expected discounted return $J^\pi$ using sampled data. However, due to the stochastic nature of Algorithm 1, we focus on analyzing the expected return $E[J^{\pi_{\theta_k}}]$ at each iteration $k$.

Note that the algorithm requires samples $s_k \sim d^{\pi_{\theta_k}}$ from the occupation measure at each iteration, which is a common assumption in most works on the discounted reward setting Zhang et al. (2020); Konda and Tsitsiklis (1999); Bhatnagar et al. (2009); Chen et al. (2021); Kumar et al. (2023); Olshevsky and Gharesifard (2023). This can be achieved by initializing the Markov chain with $s_0 \sim \mu$, and at each step $i$, continuing the chain with probability $\gamma$ by sampling $s_{i+1} \sim P^{\pi_{\theta_k}}(\cdot|s_i)$, or terminating the chain with probability $(1 - \gamma)$. Once the chain terminates, we randomly select a state uniformly as $s_k$. This process ensures that the state $s_k$ is sampled from $d^{\pi_{\theta_k}}$. However, this approach increases the average computational complexity by a factor of $\frac{1}{1-\gamma}$. There are potentially more efficient approaches to achieve this sampling, and several studies Wu et al. (2022); Xu et al. (2020b) have investigated convergence analysis using Markovian sampling. However, we omit these considerations here for simplicity.

**Assumption 3.1.** [Sufficient Exploration Assumption] There exists a $\lambda > 0$ such that:
$$\langle Q^\pi - Q, D^\pi (I - \gamma P_\pi) Q^\pi - Q \rangle \geq \lambda \|Q^\pi - Q\|_2^2,$$
where $P_\pi((s', a'), (s, a)) = P(s'|s, a)\pi(a'|s')$ and $D^\pi((s', a'), (s, a)) = \mathbf{1} \left( (s', a') = (s, a) \right) d^\pi(s)\pi(a|s)$.

Throughout this paper, we adopt the exploration assumption mentioned above, which is standard and, to the best of our knowledge, has been made in all prior works Olshevsky and Gharesifard (2023); Chen et al. (2021); Chen and Zhao (2024); Bhatnagar et al. (2009); Konda and Tsitsiklis (1999); Zhang et al. (2020). Note that the both actor and critic evolving simultaneously, with actor updating the policy with the imprecise critic's feedback (Q-value) and critic tracking the Q-value of the changing policies. This complex interdependent analysis of error is the core subject of investigation

of this paper. However, the above assumption provides the bare minimum condition that the critic convergence to the Q-value of any fixed policy in expectation.Specifically, for any fixed policy $\pi$, the Q-value update given by (line 4 of Algorithm 1):

$$Q_{m+1}(s,a) = Q_m(s,a) + \beta_k \left[ R(s,a) + \gamma Q_m(s',a') - Q_m(s,a) \right], \tag{2}$$

where $s \sim d^\pi, a \sim \pi(\cdot|s), s' \sim P(\cdot|s,a), a' \sim \pi(\cdot|s')$, $Q_m$ converges to the true Q-value $Q^\pi$ in expectation, under this exploration assumption, More precisely, $\|EQ_m - Q^\pi\| \leq c^m$ for some $c < 1$ (see Lemma A.1).The above assumption is satisfied if all the coordiantes of Q-values are updated often enough. This can be ensured by having strictly positive support of initial state-distribution on all the states $(\min_s \mu(s) > 0)$ and having sufficient exploratory policies.

**Local Convergence To Global Convergence.** Convergence of single time-scale actor-critic (Algorithm 1) has been studied for a long time, Konda and Tsitsiklis (1999); Bhatnagar et al. (2009); Zhang et al. (2020); Olshevsky and Gharesifard (2023); Chen et al. (2021); Chen and Zhao (2024). These works establish local convergence bounding the average expected square of gradient of the return, with following state-of-the-art rate

$$\sum_{k=1}^{K} \frac{1}{K} E\|\nabla J^{\pi_k}\|^2 \leq O(K^{-\frac{1}{2}}).$$

This local sample complexity of $O(\epsilon^{-2})$ translates to global sample complexity of $O(\epsilon^{-4})$, as shown in the result below.

**Proposition 3.2.** *A local $\epsilon$-close stationary policy is equivalent to an $\sqrt{\epsilon}$-close global optimal policy. That is*

$$E\|\nabla J^{\pi_{\theta_k}}\|^2 \leq O(k^{-\frac{1}{2}}) \quad \implies \quad J^* - EJ^{\pi_{\theta_k}} \leq O(k^{-\frac{1}{4}}).$$

*Proof.* The proof follows directly from Gradient Domination Lemma 2.1 and Jensen's inequality, with more details in the appendix. □

Now we present below the main result of the paper that proves the convergence of the Algorithm 1 with sample complexity of $O(\epsilon^{-3})$ to achieve $\epsilon$-close global optimal policy.

**Theorem 3.3** (Main Result). *For step size $\beta_k, \eta_k = O(k^{-\frac{2}{3}})$ in Algorithm 1, we have*

$$J^* - EJ^{\pi_{\theta_k}} \leq O(k^{-\frac{1}{3}}), \qquad \forall k \geq 0.$$

The above result significant improves upon the existing sample complexity of $O(\epsilon^{-4})$ Olshevsky and Gharesifard (2023); Chen et al. (2021); Chen and Zhao (2024) as summarized in Table 1. Additionally, the convergence is established on the last iterate in the result above. If we follow the analysis in Chen and Zhao (2024) and plug in the gradient domination condition at the end as shown in the Proposition 3.2, the convergence in value function space will be on the best iterate (in addition to having an inferior rate).

The convergence analysis consists of following three main components, discussed in details in the section next.

1. **Deriving Recursions for Actor and Critic Errors:** The first step involves formulating the recursions for the actor and critic errors, which are inherently complex and interconnected. This step is inspired by the approach outlined in Chen and Zhao (2024).

2. **Identifying a well behaved Lyapunov Term:** While prior works utilize the standard convex-optimization technique to rearrange the recursion, expressing the "norm of the gradient" through a telescoping sum to establish local convergence Chen and Zhao (2024), this work takes a novel direction. Specifically, it leverages the additional problem structure, encapsulated in the Gradient Domination Lemma, to identify a Lyapunov term—defined as the sum of the actor error and the square of the critic error—and derive a Lyapunov recursion.

3. **Developing an elegant ODE domination Method to Bound the Lyapunov Recursion:** The derived Lyapunov recursion poses significant challenges compared to the exact gradient case studied in Xiao (2022), primarily due to the presence of time-decaying learning rates. To address this, we develop an elegant ODE domination methodology that enables us to establish bounds on the Lyapunov recursion. These bounds, in turn, yield precise characterizations of both the actor and critic errors.

## 4 Convergence Analysis

In this section, we present the convergence analysis of Algorithm 1, but first, we introduce some shorthand notations for clarity. Throughout the paper, we use the following conventions:

$$J^k = J^{\pi_{\theta_k}} \in \mathbb{R}, \quad A^k = A^{\pi_{\theta_k}} \in \mathbb{R}^{\mathcal{S} \times \mathcal{A}}, \quad Q^k = Q^{\pi_{\theta_k}} \in \mathbb{R}^{\mathcal{S} \times \mathcal{A}}, \quad d^k = d^{\pi_{\theta_k}} \in \mathbb{R}^{\mathcal{S}}.$$

Additionally, we define $a_k, z_k, y_k \in \mathbb{R}$ as

- $a_k := E[J^* - J^k]$, which represents the expected sub-optimality.
- $z_k := \sqrt{E\|Q_k - Q^k\|^2}$, which denotes the expected critic tracking error.
- $y_k := \sqrt{E\|\nabla J^k\|^2}$, which denotes the expected norm of the gradient.

We summarize all the useful constants in the Table 4. We begin by deriving an actor recursion, which is essentially a sufficient increase lemma for our noisy and biased gradient ascent (Line 3 of Algorithm 1). This recursion arises from the smoothness properties of the return and serves as an extension of its non-noisy version presented in Mei et al. (2022).

**Lemma 4.1.** *[Actor Recursion] Let $\theta_k$ be the iterates from Algorithm 1, then the sub-optimality decreases as*

$$a_{k+1} \leq a_k - c_1 \eta_k y_k^2 + c_2 \eta_k y_k z_k + c_3 \eta_k^2.$$

The recursion above illustrates the dependence of sub-optimality progression on various terms. The second term, $\frac{\eta_k y_k^2}{1-\gamma}$, indicates that the sub-optimality decreases proportionally to the square of the gradient norm and the learning rate, which is consistent with the expected behavior of gradient ascent on a smooth function in standard optimization. The term $\frac{2\eta_k y_k z_k}{1-\gamma}$ represents the bias arising from the error in Q-value estimation (critic error), implying that higher critic estimation error reduces the improvement in the policy. Finally, the term $\frac{2L\eta_k^2}{(1-\gamma)^4}$ accounts for the variance (second moment) of the updates.

Now, we shift our focus to the critic error. The exploration Assumption 3.1 ensures the evaluation of the policy (Q-value estimation in expectation) through samples with respect to a fixed policy. However, in Algorithm 1, the policy changes at every iteration, which makes the derivation of the result below somewhat more challenging.

**Lemma 4.2.** *[Critic Recursion] In Algorithm 1, critic error follows the following recursion*

$$z_{k+1}^2 \leq (1 - c_4 \beta_k) z_k^2 + c_5 \beta_k^2 + c_6 \eta_k^2 + c_7 \eta_k y_k z_k,$$

*where constants $c_i$ are defined in the appendix.*

The term $(1 - c_4 \beta_k) z_k^2$ represents the geometric decrease of the critic error, as the Q-value is a contraction operator. The terms $c_5 \beta_k^2$ and $c_6 \eta_k^2$ arise from the variance in the critic and policy updates. Finally, the term $c_7 \eta_k y_k z_k$ reflects the effect of the "moving goalpost," where the critic evaluates a policy that changes in each iteration by an amount proportional to $y_k$.

**Lemma 4.3** (Gradient Domination). *The sub-optimality is upper bound by gradient as*

$$a_k \leq c_8 y_k.$$

The result above upper bounds the sub-optimality with the gradient, which follows Lemma 2.1 and Jensen's inequality. In summary, we have the following set of simplified recursions:

$$\textbf{Actor:} \quad a_{k+1} \leq a_k - c_1 \eta_k y_k^2 + c_2 \eta_k y_k z_k + c_3 \eta_k^2 \tag{3}$$
$$\textbf{Critic:} \quad z_{k+1}^2 \leq z_k^2 - c_4 \beta_k z_k^2 + c_5 \beta_k^2 + c_6 \eta_k^2 + c_7 \eta_k y_k z_k$$
$$\textbf{GDL:} \quad a_k \leq c_8 y_k.$$

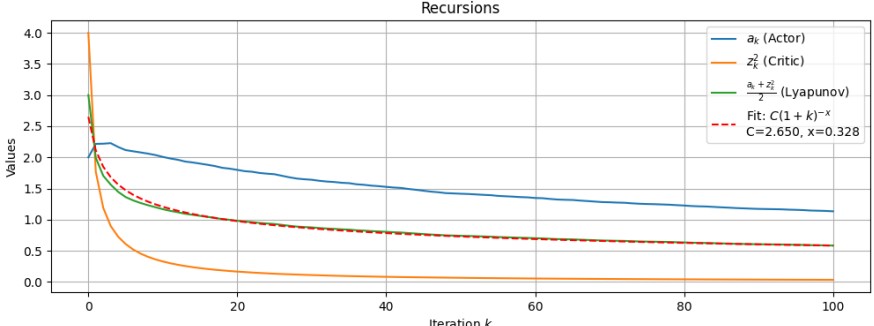

Figure 1: Actor-Critic recursion in (3): Random $c_i$, $10\eta_k = \beta_k = (1+k)^{-\frac{2}{3}}$, $a_0, z_0 = 2$.

Solving these interdependent recursions is highly challenging and forms the core technical contribution of this paper. Notably, we cannot guarantee a monotonic decrease in either the sub-optimality $a_k$ or the critic error $z_k$ across iterations, since $a_{k+1}$ tends to decrease while $z_{k+1}^2$ increases with the growth of $y_k$. A crucial observation is that the Lyapunov term $x_{k+1} := a_{k+1} + z_{k+1}^2$ exhibits a consistent decrease as $y_k$ increases, as shown in Figure 1. This highlights the stability and utility of the Lyapunov term in characterizing the system's behavior. Now to formally prove this, we combine the actor and critic recursions, assume $\beta_k = c_\beta \eta_k$, and apply additional algebraic manipulations (detailed in the appendix). This leads to the following recursion:

$$a_{k+1} + z_{k+1}^2 \le a_k + z_k^2 - c_{12}\eta_k\left(y_k + z_k^2\right)^2 + c_{11}\eta_k^2.$$

Using the Gradient Domination Lemma (GDL), we derive the Lyapunov recursion:

$$x_{k+1} \le x_k - c_{13}\eta_k x_k^2 + c_{11}\eta_k^2,$$

which can be solved as stated in the following result.

**Lemma 4.4** (ODE Domination Lemma). *Given $\eta_k = c_{14}\left(\frac{1}{\frac{1}{x_0^3}+c_{15}k}\right)^{\frac{2}{3}}$, the recursion $x_{k+1} \le x_k - c_{13}\eta_k x_k^2 + c_{11}\eta_k^2$ satisfies the bound:*

$$x_k \le \left(\frac{1}{\frac{1}{x_0^3} + c_{15}k}\right)^{\frac{1}{3}},$$

*Proof.* The detailed steps of the proof are provided in the appendix. The key idea in solving the recursion is to establish that $x_k$ lies below the trajectory of the following ODE:

$$\frac{du_k}{dk} = -c_{13}\eta_k u_k^2 + c_{11}\eta_k^2.$$

We simplify this by appropriately choosing $\eta_k = c_{14}u_k^2$, leading to the reduced ODE: $\frac{du_k}{dk} = -c_{15}u_k^4$, whose solution is: $u_k = \left(\frac{1}{\frac{1}{u_0^3} + c_{15}k}\right)^{\frac{1}{3}}$. $\qquad\square$

Using the above result, we conclude that $a_k = O(k^{-\frac{1}{3}})$ and $\eta_k, \beta_k = O(k^{-\frac{2}{3}})$, thus completing the convergence analysis. Although, we retrospectively chose the best learning rates $\beta_k, \eta_k = O(k^{-\frac{2}{3}})$ for the presentation simplifications. But we have developed a general framework in the appendix that gives the rates for different possible step-sizes schedules.

Additionally, the result below shows that our critic error follows $z_k = O(k^{-\frac{1}{3}})$, as compared to the $O(k^{-\frac{1}{4}})$ rate achieved in Chen and Zhao (2024).

**Corollary 4.5.** *The critic error decreases similar to the actor error as*

$$z_k \leq \left( \frac{1}{c_{16} + c_{17}k} \right)^{\frac{1}{3}}.$$

*Proof.* From Lemma 4.2, we have

$$z_{k+1}^2 \leq (1 - c_4\beta_k)z_k^2 + c_5\beta_k^2 + c_6\eta_k^2 + c_7\eta_k y_k z_k. \tag{4}$$

$\square$

| Constant | Definition | Remark |
|----------|------------|--------|
| $J^k \in \mathbb{R}$ | $J^{\pi_{\theta_k}}$ | Return at iterate $k$ |
| $A^k \in \mathbb{R}^{\mathcal{S} \times \mathcal{A}}$ | $A^{\pi_{\theta_k}}$ | Advantage value at iterate $k$ |
| $Q^k \in \mathbb{R}^{\mathcal{S} \times \mathcal{A}}$ | $Q^{\pi_{\theta_k}}$ | Q-value at iterate $k$ |
| $d^k \in \mathbb{R}^{\mathcal{S}}$ | $d^{\pi_{\theta_k}}$ | Occupation measure at iterate $k$ |
| $a_k \in \mathbb{R}$ | $E[J^* - J^k]$ | Sub-optimality at iterate $k$ |
| $z_k \in \mathbb{R}$ | $\sqrt{E\|Q_k - Q^k\|}$ | Critic mean squared error at iterate $k$ |
| $y_k \in \mathbb{R}$ | $\sqrt{E\|\nabla J^k\|^2}$ | Expected squared norm of the return at iterate $k$ |
| $x_k \in \mathbb{R}$ | $a_k + z_k^2$ | Lyapunov value at iterate $k$ |
| $u_k \in \mathbb{R}$ | $\left( \frac{1}{\frac{1}{u_0^3} + c_{15}k} \right)^{\frac{1}{3}}$ | Solution to the ODE $\frac{du_k}{dk} = -c_{15}u_k^4$ |
| $c_i \in \mathbb{R}$ | Place holder constants | See appendix |

Table 2: Definitions of cseful constants: Iterate $k$ is generated from Algorithm 1

# 5    Discussion

We establish the global convergence of actor-critic algorithms with a significantly improved sample complexity of $O(\epsilon^{-3})$ for obtaining $\epsilon$-close global optimal policy, compared to the existing rate of $O(\epsilon^{-4})$ derived from $O(\epsilon^{-2})$ complexity for $\epsilon$-close stationary policy Chen and Zhao (2024). This brings us closer to the lower bound complexity of $O(\epsilon^{-2})$ for reinforcement learning Auer et al. (2008). The framework we propose is quite general and could potentially be extended to other settings, such as average reward, function approximation, or Markovian noise. We leave these extensions for future work.

Moreover, this framework for addressing the two-time-scale coupling, combined with our novel and elegant methodology for bounding the recursions, can serve as a foundation for analyzing other two-time-scale algorithms.

**Can we improve the complexity further?**    Our work proposes a learning rate schedule for both the critic and actor, decaying as $k^{-\frac{2}{3}}$ with iteration $k$, which we believe through our investigation, achieves the optimal sample complexity of $O(\epsilon^{-3})$ that these recursions can possible yield. Consequently, we need to shift our approach in deriving these recursions for improvement in the sample complexity. All prior approaches, including our own, focus on bounding the variance of the critic error $\sqrt{E\|Q^k - Q_k\|^2}$. However, for the analysis of the actor's recursion, it suffices to bound the bias $\|Q^k - \mathbb{E}Q_k\|$. Through careful investigation, we have come to believe that our current analysis, which relies on variance bounds, has reached the best possible sample complexity limit of $O(\epsilon^{-3})$.

In contrast, an analysis based on bias has the potential to achieve further improvements, possibly reducing the complexity to the theoretical lower bound of $O(\epsilon^{-2})$.

A key insight lies in the fundamental difference between variance and bias: even for a fixed policy, variance remains non-zero, whereas bias vanishes. Specifically, current variance-based approaches necessitate diminishing learning rates for both the actor and the critic to ensure decreasing variance. In contrast, the bias term can tend to zero even with a constant critic learning rate, requiring only a diminishing learning rate for the actor. This observation suggests that focusing on bias may be a more promising direction, but it also presents significant analytical challenges that remain unexplored.

In summary, we hypothesize that the current sample complexity of $O(\epsilon^{-3})$ could be improved to $O(\epsilon^{-2})$ by focusing on bias rather than variance. This shift in focus may allow for a constant (or very slowly decaying) critic step size, only requiring diminishing actor step size. In addition, we believe our new methodology for solving recursions may play a crucial role in unlocking these new research directions and opportunities.

**Extension to continuous spaces.** Our analysis is limited to the tabular setting and does not extend to large or continuous state spaces (e.g., robotics) due to the $\sqrt{S}$ dependence in the Gradient Dominant Lemma (GDL) 2.1. Intuitively, $\sqrt{S}$ reflects the diameter of the policy space ($\max_{\pi,\pi'}\|\pi - \pi'\|_2$), which could be replaced by parameter space diameter ($\max_{\theta,\theta'}\|\theta - \theta'\|_2$), in function approximation. This extencion directly enables the non-tabular versions of the exact gradient convergence results in Xiao (2022); Mei et al. (2022) and consequently our actor–critic complexity results, with minor modifications in the current analysis.

**Using multiple samples for critic estimation.** While many double-loop actor–critic methods use (too) many critic samples per actor update, our work takes the opposite extreme—using only one. Exploring whether an optimal trade-off exists between these two extremes is an interesting future direction.

**Re-use of samples.** We believe that re-using samples could reduce the total number of new samples needed to below $O(\epsilon^{-3})$. This direction is particularly interesting for bridging offline and online RL, which we leave for future work.

## Acknowledgments and Disclosure of Funding

This research was partially supported by Israel PBC- VATAT, by the Technion Artificial Intelligent Hub (Tech.AI) and by the Israel Science Foundation (grant No. 447/20).

Additionally, part of this work was supported by the Israel Science Foundation (grant No. 3019/24).

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

# Contents

# A   Supporting Results

## A.1   On Sufficient Exploration Assumption 3.1

**Lemma A.1.** *Under the Assumption 3.1, the update rule* (2)*, converges as*

$$\|\mathbb{E}Q_k - Q^\pi\|_2 \to \alpha^k \|\mathbb{E}Q_0 - Q^\pi\|_2,$$

*where* $\alpha = \sqrt{1 - \frac{\lambda^2}{2}}$ *taking* $\beta_k = \frac{\lambda}{2}$.

*Proof.* From Proposition A.3, we have $\|EQ_{k+1} - Q^\pi\| \leq \alpha\|EQ_{k+1} - Q^\pi\|$, from which the result follows. $\qquad\square$

We define $P_\pi((s', a'), (s, a)) = P(s'|s, a)\pi(a'|s')$ and $D^\pi((s', a'), (s, a)) = \mathbf{1}\left((s', a') = (s, a)\right)$ $(1 - \gamma)\sum_{n=0}^{\infty}\gamma^n\mu^T(P^\pi)^n(s)$.

**Proposition A.2.** $c_\gamma = \max_{\pi, Q}\frac{\|D^\pi(I - \gamma P_\pi)Q\|}{\|Q\|} \leq 1 + \gamma$.

*Proof.*

$$\|D^\pi(I - \gamma P_\pi)Q\| \le \|D^\pi Q\| + \gamma\|D^\pi P_\pi Q\| \tag{5}$$

$$\le \|Q\| + \gamma\|D^\pi P_\pi Q\|, \qquad (\text{as } \sum_{s,a}|D((s,a),(s,a))| = 1) \tag{6}$$

$$= \|Q\| + \gamma\sqrt{\sum_{s,a}\big(\,d(s,a)\langle P_\pi(\cdot|(s,a)), Q\rangle\,\big)^2}, \tag{7}$$

$$\le \|Q\| + \gamma\sqrt{\sum_{s,a}\big(\,d(s,a)\|P_\pi(\cdot|(s,a))\|\|Q\|\,\big)^2}, \tag{8}$$

$$\le \|Q\| + \gamma\|Q\|\sqrt{\sum_{s,a}\big(\,d(s,a)\,\big)^2\|P_\pi(\cdot|(s,a))\|^2}, \tag{9}$$

$$\le \|Q\| + \gamma\|Q\|\sqrt{\sum_{s,a}d(s,a)\|P_\pi(\cdot|(s,a))\|_1^2}, \tag{10}$$

$$= (1 + \gamma)\|Q\|. \tag{11}$$

$\square$

**Proposition A.3.** *For any policy $\pi$, given $T_\beta^\pi Q = Q + \beta D^\pi \Big[ R + \gamma P_\pi Q - Q \Big]$, we have*

$$\|Q^\pi - T_\beta^\pi Q\| \le \sqrt{1 - \frac{\lambda^2}{2}}\|Q^\pi - Q\|_2.$$

*Proof.*

$$U := D^\pi \Big[ R - (I - \gamma P_\pi)Q \Big] \tag{12}$$

$$= D^\pi \Big[ Q^\pi - \gamma P_\pi Q^\pi - (I - \gamma P_\pi)Q \Big], \qquad (\text{using } Q^\pi = R + \gamma P_\pi Q^\pi) \tag{13}$$

$$= D^\pi \big( I - \gamma P_\pi \big)\big( Q^\pi - Q \big) \tag{14}$$

Lets look at

$$\|Q^\pi - T_\beta^\pi Q\|^2 = \|Q^\pi - Q - \beta U\|^2, \qquad (\text{definition of } T_\beta^\pi Q = Q + \beta U)$$
$$= \|Q^\pi - Q\|^2 + \beta^2\|U\|^2 - 2\beta\langle Q^\pi - Q, U\rangle$$
$$\le \|Q^\pi - Q\|^2 + \beta^2\|U\|^2 - 2\beta\lambda\|Q^\pi - Q\|^2, \qquad (\text{from Assumption 3.1})$$
$$\le (1 + 2\beta^2 - 2\beta\lambda)\|Q^\pi - Q\|_2^2, \qquad (\text{from Proposition A.2})$$
$$\le (1 - \frac{\lambda^2}{2})\|Q^\pi - Q\|_2^2, \qquad (\text{taking } \beta = \frac{\lambda}{2}).$$

$\square$

## A.2 Local Convergence to Global Convergence: Proof of Proposition 3.2

**Proposition A.4.** *If $E\|\nabla J^k\|_2^2 \le O(k^{-\frac{1}{2}})$ then $J^* - EJ^{\pi_k} \le O(k^{-\frac{1}{4}})$.*

*Proof.* From Gradient Domination Lemma 2.1 and Jensen's inequality, we have

$$E\|\nabla J^k\|_2^2 \ge E\Big[ J^* - J^k \Big] \ge \frac{c^2}{SC_{PL}^2}\Big[ J^* - EJ^k \Big]^2.$$

Hence if $E\|\nabla J^{\pi_k}\|_2^2 \le O(k^{-\frac{1}{2}})$ then $\Big[ J^* - EJ^{\pi_k} \Big]^2 \le O(k^{-\frac{1}{2}})$, implying $J^* - EJ^k \le O(k^{-\frac{1}{4}})$.

$\square$

# B Deriving Recursions

**Notations.** Recall that $J^k = J^{\pi_{\theta_k}}, A^k = A^{\pi_{\theta_k}}, Q^k = Q^{\pi_{\theta_k}}, d^k = d^{\pi_{\theta_k}}, a_k = E[J^* - J^k], y_k = \sqrt{E\|\nabla J^k\|^2}, z_k = \sqrt{E\|Q_k - Q^k\|^2}$ are used as shorthands. Further $Q_k, A_k$ are iterates from Algorithm 1, and $\mathbf{1}_k \in \{0,1\}^{\mathcal{S} \times \mathcal{A}}$ is indicator for $(s_k, a_k)$ in the Algorithm 1. We refer Hadamard product by $\odot$, defined as $(a \odot b)(i) = a(i)b(i)$.

| Constant | Definition | Remark |
|---|---|---|
| $\lambda$ | | Sufficient Exploration constant |
| $L$ | $\frac{8}{(1-\gamma)^3}$ | Smoothness constant |
| $c_g$ | $\frac{\sqrt{S}C_{PL}}{c}$ | GDL constant |
| $L_1^\pi = 2$ | $\|\pi_{\theta_{k+1}} - \pi_{\theta_k}\| \le L_1^\pi \|\theta_{k+1} - \theta_k\|$ | Lipschitz constant of policy w.r.t. $\theta$ |
| $c_q = \frac{2\sqrt{S}A}{(1-\gamma)^4}$ | $\|Q^k - Q^{k+1}\| \le c_q \eta_k$ | Lipschitz constant, see Proposition B.3 |
| $c_u \le \frac{3}{1-\gamma}$ | $\|U_k\| \le c_u$ | Proposition B.1 |
| $L_2^q = \frac{8\sqrt{S}A}{(1-\gamma)^3}$ | $\|Q^k - Q^{k+1} + \nabla Q^k(\theta_{k+1} - \theta_k)\| \le \frac{1}{2}L_2^q\|\theta_{k+1} - \theta_k\|^2$ | Smoothness of $Q$, see Proposition B.6 |
| $c_z = \frac{2\sqrt{S}A}{(1-\gamma)}$ | $\max_k\|Q_k - Q^k\| \le c_z$ | Upper bound on $z_k$, see Proposition B.5 |
| $c_\beta = \frac{\beta_k}{\eta_k}$ | $\frac{9SA^2}{2(1-\gamma)^5}$ | Actor-critic scale ratio |
| $c_\eta$ | $2c_u^2 c_\beta^2 + \frac{4L}{(1-\gamma)^4} + 2c_q^2 + \frac{2L_2^q c_z}{(1-\gamma)^4}$ | |
| | $\le \frac{818S^2A^4}{(1-\gamma)^{12}}$ | |
| $c_l$ | $\frac{1}{4}\min\{\frac{1}{c_g^2 1-\gamma)}, \frac{2\lambda c_\beta}{c_z^2}\}$ $= \frac{1}{4(1-\gamma)}\min\{\frac{c^2}{SC_{PL}^2}, \frac{9\lambda S}{4(1-\gamma)^2}\}$ | ODE constant |

Table 3: Constants

In this section, we derive the following recursions:

$$a_{k+1} \le a_k - \frac{\eta_k}{1-\gamma}y_k^2 + \frac{2\eta_k}{1-\gamma}y_k z_k + \frac{4L\eta_k^2}{(1-\gamma)^4}$$

$$a_k \le c_g y_k$$

$$z_{k+1}^2 \le (1 - 2\lambda\beta_k)z_k^2 + 2c_u^2\beta_k^2 + 2c_q^2\eta_k^2 + \frac{2L_2^q}{(1-\gamma)^4}\eta_k^2 z_k + \frac{2\gamma\sqrt{S}A}{(1-\gamma)^3}\eta_k y_k z_k,$$

where the constants are described in Table 3.

## B.1 Useful Constants

The constants appears in upcoming sub-sections while deriving and solving the recursions. Reader may skip and come back to this subsection later.

**Proposition B.1.**

$$c_u := \max_{\|Q\|_\infty \le \frac{1}{1-\gamma}, s,s'\in\mathcal{S}, a,a'\in A}\left[R(s,a) + \gamma Q_k(s',a') - Q_k(s,a)\right] \le \frac{3}{1-\gamma}.$$

*Proof.*

$$|R(s,a) + \gamma Q_k(s',a') - Q_k(s,a)| \le |R(s,a)| + \gamma |Q_k(s',a')| + |Q_k(s,a)| \tag{15}$$

$$\le 1 + \frac{2}{1-\gamma} \le \frac{3}{1-\gamma}. \tag{16}$$

$\square$

**Proposition B.2.** *[Lipschitz constant of value function]*

$$\|v^{k+1} - v^k\|_\infty \le \frac{\sqrt{A}}{(1-\gamma)^4}\eta_k.$$

*Proof.*

$$\|v^\pi - v^{\pi'}\|_\infty \le \|(I - \gamma P^\pi)^{-1}[(R^\pi - R^{\pi'}) + \gamma(P^\pi - P^{\pi'})v^{\pi'}]\|_\infty \tag{17}$$

$$\le \tfrac{1}{1-\gamma}\big(\|R^\pi - R^{\pi'}\|_\infty + \gamma\|(P^\pi - P^{\pi'})v^{\pi'}\|_\infty\big). \tag{18}$$

$$\le \frac{1}{1-\gamma}\max_s \left[ \|\pi_s' - \pi_s\|_1 + \gamma\frac{\|\pi_s' - \pi_s\|_1}{1-\gamma} \right] \tag{19}$$

$$\le \frac{\|\pi_s' - \pi_s\|_1}{(1-\gamma)^2} \tag{20}$$

$$\le \max_s \frac{\sqrt{A}}{2(1-\gamma)^2}\|\theta'(s) - \theta(s)\|_1, \qquad (\text{as } \|\pi(s)_{\theta'} - \pi(s)_\theta\|_1 \le \frac{\sqrt{A}}{2}\|\theta'(s) - \theta(s)\|_1). \tag{21}$$

Hence

$$\|v^{k+1} - v^k\|_\infty \le \frac{\sqrt{A}}{(1-\gamma)^4}\eta_k,$$

as $\|\theta_{k+1} - \theta_k\| = \frac{2\eta_k}{(1-\gamma)^2}$. $\square$

**Proposition B.3.**

$$\|Q^k - Q^{k+1}\| \le c_q\eta_k,$$

*where* $c_q = \frac{2\sqrt{SA}}{(1-\gamma)^4}$

**Proposition B.4.**

$$\|Q^k - Q^{k+1}\|_\infty = \|R + \gamma Pv^k - R - \gamma Pv^{k+1}\|_\infty \tag{22}$$

$$= \gamma\|Pv^k - Pv^{k+1}\|_\infty \tag{23}$$

$$= \gamma\|v^k - v^{k+1}\|_\infty \tag{24}$$

$$\le \gamma\frac{\sqrt{A}\eta_k}{(1-\gamma)^4}, \qquad \textit{(from Proposition B.2).} \tag{25}$$

**Proposition B.5.**

$$\max_k\|Q_k - Q^k\| \le c_z,$$

*where* $c_z = \frac{2\sqrt{SA}}{(1-\gamma)}$

*Proof.*

$$\|Q_k - Q^k\| \le \sqrt{SA}\|Q_k - Q^k\|_\infty \le \frac{2\sqrt{SA}}{(1-\gamma)}. \tag{26}$$

$\square$

**Proposition B.6.**

$$|\frac{d^2 Q^{\pi_\theta}}{d\alpha^2}| \leq \frac{8\gamma}{(1-\gamma)^3}$$

*and*

$$\|Q^k - Q^{k+1} + \nabla Q^k(\theta_{k+1} - \theta_k)\| \leq \frac{4\sqrt{SA}}{(1-\gamma)^3}\|\theta_{k+1} - \theta_k\|^2.$$

*Proof.*

$$|\frac{d^2 Q^{\pi_\theta}}{d\alpha^2}| = |\frac{d^2}{d\alpha^2}\left[ R(s,a) + \gamma \sum_{s'} P(s'|s,a) v^{\pi_\theta}(s') \right]| \tag{27}$$

$$\leq \gamma \sum_{s'} P(s'|s,a)|\frac{d^2}{d\alpha^2} v^{\pi_\theta}(s')| \tag{28}$$

$$= \gamma L. \tag{29}$$

Hence,

$$\|Q^k - Q^{k+1} + \nabla Q^k(\theta_{k+1} - \theta_k)\| \leq \sqrt{SA}\|Q^k - Q^{k+1} + \nabla Q^k(\theta_{k+1} - \theta_k)\|_\infty \tag{30}$$

$$\leq \sqrt{SA}\frac{\gamma L}{2}\|\theta_{k+1} - \theta_k\|^2 \tag{31}$$

$$\leq \frac{4\sqrt{SA}}{(1-\gamma)^3}\|\theta_{k+1} - \theta_k\|^2. \tag{32}$$

$\square$

**Proposition B.7.**

$$c_\eta \leq \frac{818 S^2 A^4}{(1-\gamma)^{12}}$$

*Proof.* From definition, we have

$$c_\eta = 2c_u^2 c_\beta^2 + \frac{4L}{(1-\gamma)^4} + 2c_q^2 + \frac{2L_2^q c_z}{(1-\gamma)^4} \tag{33}$$

$$\leq \frac{729 S^2 A^4}{2(1-\gamma)^{12}} + \frac{32}{(1-\gamma)^7} + \frac{8SA^2}{(1-\gamma)^8} + \frac{32SA}{(1-\gamma)^8}, \qquad (\text{as } L = \frac{8}{(1-\gamma)^3}) \tag{34}$$

$$\leq \frac{818 S^2 A^4}{(1-\gamma)^{12}}. \tag{35}$$

$\square$

## B.2  Actor Recursion: Proof of Lemma 4.1

**Lemma B.8** (Sufficient Increase Lemma). *Let $\theta_k$ be the iterate obtained Algorithm 1. Then,*

$$E[J^{k+1} - J^k] \geq \frac{\eta_k}{1-\gamma}E\left[ \|\nabla J^k\|^2 + \langle \nabla J^k, d^k \odot (A_k - A^k)\rangle - \frac{2L\eta_k}{(1-\gamma)^3} \right],$$

*where $L = \frac{8}{(1-\gamma)^3}$.*

*Proof.* From the smoothness of the return, we have

$$E\left[ J^{k+1} - J^k \right] \geq E\left[ \langle \nabla J^k, \theta_{k+1} - \theta_k \rangle - \frac{L}{2}\|\theta_{k+1} - \theta_k\|^2 \right],$$

$$\geq E\left[ \frac{\eta_k}{1-\gamma}\langle \nabla J^k, A_k \odot \mathbf{1}_k \rangle - \frac{L\eta_k^2}{2(1-\gamma)^2}A_k^2 \mathbf{1}_k \right], \qquad (\text{from update rule in Algorithm 1}$$

$$\geq \frac{\eta_k}{1-\gamma}E\left[ \langle \nabla J^k, d^k \odot A_k \rangle - \frac{2L\eta_k}{(1-\gamma)^3} \right], \qquad (\text{ as } (s_k, a_k) \sim d^k\odot \text{ and } \|A_k\|_\infty \leq \frac{2}{1-\gamma})$$

$$\geq \frac{\eta_k}{1-\gamma}E\left[ \|\nabla J^k\|_2^2 + \langle \nabla J^k, d^k \odot (A_k - A^k)\rangle - \frac{2L\eta_k}{(1-\gamma)^3} \right], \qquad (\text{ as } \nabla J^k = d^k \odot A^k).$$

$\square$

**Proposition B.9.** *We have*

$$E \left| \langle \nabla J^k, d^k \odot (A_k - A^k) \rangle \right| \leq 2\sqrt{E\|\nabla J^k\|^2}\sqrt{E\|Q_k - Q^k\|^2}.$$

*Proof.* We have

$$\left| \langle \nabla J^k, d^k \odot (A_k - A^k) \rangle \right| \leq \|\nabla J^k\| \|d^k \odot (A_k - A^k)\|, \qquad \text{(from Cauchy inequlaity)} \quad (36)$$

$$\leq \|\nabla J^k\| \|d^k\| \|(A_k - A^k)\|_\infty, \qquad \text{(as } \sum_i (a_i b_i)^2 \leq (\max_i a_i^2)(\sum_i b_i^2)) \quad (37)$$

$$\leq \|\nabla J^k\| \|A_k - A^k\|_\infty, \qquad (\text{ as } 1 = \|d^k\|_1 \geq \|d^k\|_2) \quad (38)$$

$$(39)$$

Additionally, from definition, we have

$$|A_k(s,a) - A^k(s,a)| = |Q_k(s,a) - \sum_a \pi(a|s)Q_k(s,a) - Q^k(s,a) + \sum_a \pi(a|s)Q_k(s,a)| \quad (40)$$

$$\leq |Q_k(s,a) - Q^k(s,a)| + |\sum_a \pi(a|s)Q_k(s,a) - \sum_a \pi(a|s)Q_k(s,a)|, \qquad \text{(Triangle inequality)}$$

$$(41)$$

$$\leq \|Q_k - Q^k\|_\infty + \sum_a \pi(a|s)|Q_k(s,a) - Q_k(s,a)|, \quad (42)$$

$$\leq 2\|Q_k - Q^k\|_\infty. \quad (43)$$

Putting this back, we get

$$E \left| \langle d^k \odot A^k, d^k \odot (A_k - A^k) \rangle \right| \leq 2E \left[ \|\nabla J^k\| \|Q_k - Q^k\|_\infty \right], \quad (44)$$

$$\leq 2E \left[ \|\nabla J^k\| \|Q_k - Q^k\| \right], \qquad \text{(as } \|x\|_2 \geq \|x\|_\infty) \quad (45)$$

$$\leq 2\sqrt{E\|\nabla J^k\|_2^2}\sqrt{E\|Q_k - Q^k\|_2^2}, \qquad \text{(from Cauchy } (E\langle x,y\rangle)^2 \leq E\|x\|^2 E\|y\|^2). \quad (46)$$

$$\square$$

**Lemma B.10.** *[Actor Recursion] We have*

$$a_k - a_{k+1} \geq \frac{\eta_k}{1-\gamma} \left[ y_k^2 - 2y_k z_k - \frac{2L\eta_k}{(1-\gamma)^3} \right],$$

*where $L = \frac{8}{(1-\gamma)^3}$.*

*Proof.* From Sufficient Increase Lemma B.8, we have

$$E[J^{k+1} - J^k] \geq \frac{\eta_k}{1-\gamma} E \left[ \|\nabla J^k\|^2 + \langle \nabla J^k, d^k \odot (A_k - A^k) \rangle - \frac{2L\eta_k}{(1-\gamma)^3} \right],$$

$$\geq \frac{\eta_k}{1-\gamma} \left[ E\|\nabla J^k\|^2 - E|\langle \nabla J^k, d^k \odot (A_k - A^k) \rangle| - \frac{2L\eta_k}{(1-\gamma)^3} \right], \qquad \text{(as } E[a] \geq -E[|a|])$$

$$\geq \frac{\eta_k}{1-\gamma} \left[ E\|\nabla J^k\|^2 - 2\sqrt{E\|\nabla J^k\|^2}\sqrt{E\|Q_k - Q^k\|^2} - \frac{2L\eta_k}{(1-\gamma)^3} \right], \qquad \text{(from Lemma B.9).}$$

$$\square$$

## B.3 GDL Recursion: Proof of Lemma 4.3

**Proposition B.11.** *[Gradient Domination] We have*

$$a_k \leq \frac{\sqrt{S}C_{PL}}{c} y_k.$$

*Proof.* From GDL, we have

$$J^* - J^k \leq \frac{\sqrt{S}C_{PL}}{c}\|\nabla J^k\| \tag{47}$$

$$\implies E[J^* - J^k] \leq \frac{\sqrt{S}C_{PL}}{c}E\|\nabla J^k\| \tag{48}$$

$$\leq \frac{\sqrt{S}C_{PL}}{c}\sqrt{E\|\nabla J^k\|^2}, \tag{49}$$

where the last inequality comes from the Jensen's inequality $(E[x])^2 \leq E[x^2]$. □

## B.4 Critic Recursion: Proof of Lemma 4.2

Recall that in the Algorithm 1, we have the following updates: $(s, a) \sim d^k$ $s' \sim P^k(\cdot|s, a), a' \sim \pi_k(\cdot|s')$, and

$$Q_{k+1}(s, a) = Q_k(s, a) + \beta_k U_{k+1},$$

where $\|\pi_{k+1} - \pi_k\| \leq \frac{2L_1^\pi}{(1-\gamma)^2}\eta_k, \eta_k \to 0$, and $U_{k+1} = \Big[ R(s, a) + \gamma Q_k(s', a') - Q_k(s, a) \Big]$.

**Lemma B.12** (Critic Recursion). *In Algorithm 1, the critic error follows the following recursion*

$$z_{k+1}^2 \leq (1 - 2\lambda\beta_k)z_k^2 + 2c_u^2\beta_k^2 + 2c_q^2\eta_k^2 + \frac{2L_2^q}{(1-\gamma)^4}\eta_k^2 z_k + \frac{2\gamma\sqrt{S}A}{(1-\gamma)^3}\eta_k y_k z_k.$$

*Proof.* We have

$$E\|Q_{k+1} - Q^{k+1}\|^2 = E \Big\| Q_k + \beta_k U_{k+1} - Q^{k+1} \Big\|^2, \quad \text{(from update rule of } Q_k)$$

$$=E \Big\| Q_k - Q^k + \beta_k U_{k+1} + Q^k - Q^{k+1} \Big\|^2, \quad \text{(plus-minus } Q^k)$$

$$=E \Big( \|Q_k - Q^k\|^2 + \beta_k^2\|U_{k+1}\|^2 + \|Q^k - Q^{k+1}\|^2 + 2\beta_k\langle U_{k+1}, Q^k - Q^{k+1}\rangle$$
$$+ 2\beta_k\langle Q_k - Q^k, U_{k+1}\rangle + 2\langle Q_k - Q^k, Q^k - Q^{k+1}\rangle \Big), \quad \text{(expansion of } (a + b + c)^2)$$

$$\leq E \Big( (1 - 2\beta_k\lambda)\|Q_k - Q^k\|^2 + \beta_k^2\|U_{k+1}\|^2 + \|Q^k - Q^{k+1}\|^2 + 2\beta_k\langle U_{k+1}, Q^k - Q^{k+1}\rangle$$
$$+ 2\langle Q_k - Q^k, Q^k - Q^{k+1}\rangle \Big), \quad \text{(using sufficient exploration assumption)}$$

$$\leq E \Big( (1 - 2\beta_k\lambda)\|Q_k - Q^k\|^2 + 2\beta_k^2\|U_{k+1}\|^2 + 2\|Q^k - Q^{k+1}\|^2 + 2\langle Q_k - Q^k, Q^k - Q^{k+1}\rangle \Big),$$
$$\text{(using } \|a\|^2 + \|b\|^2 \geq 2\langle a, b\rangle)$$

$$\leq E \Big( (1 - 2\beta_k\lambda)\|Q_k - Q^k\|^2 + 2\beta_k^2 c_u^2 + 2\eta_k^2 c_q^2 + 2\langle Q_k - Q^k, Q^k - Q^{k+1}\rangle \Big),$$
$$\text{( as } \|Q^k - Q^{k+1}\| \leq c_q\eta_k \text{ and } \|U_k\| \leq c_u \text{ )}$$

$$\leq E \Big( (1 - 2\beta_k\lambda)\|Q_k - Q^k\|^2 + \frac{18\beta_k^2}{(1-\gamma)^2} + 2\eta_k^2 c_q^2 + 2\langle Q_k - Q^k, Q^k - Q^{k+1}\rangle \Big),$$
$$\text{(from Proposition B.1 )}$$

Now, we only focus on

$$
\begin{aligned}
&E\langle Q_k - Q^k, Q^k - Q^{k+1}\rangle \\
&\leq E\langle Q_k - Q^k, Q^k - Q^{k+1} + \nabla Q^k(\theta_{k+1} - \theta_k)\rangle + E\langle Q_k - Q^k, \nabla Q^k(\theta_{k+1} - \theta_k)\rangle, \quad \text{(plus-minus )} \\
&\leq E\Big[ \|Q_k - Q^k\|\|Q^k - Q^{k+1} + \nabla Q^k(\theta_{k+1} - \theta_k)\| + \langle Q_k - Q^k, \nabla Q^k(\theta_{k+1} - \theta_k)\rangle \Big], \quad \text{(Cauchy Schwartz )} \\
&\leq E\Big[ \frac{1}{2}L_2^q\|Q_k - Q^k\|\|\theta_{k+1} - \theta_k\|^2 + \langle Q_k - Q^k, \nabla Q^k(\theta_{k+1} - \theta_k)\rangle \Big], \quad \text{(smoothness of } Q^\pi \text{, see Table 3 )} \\
&\leq E\Big[ \frac{2L_2^q\eta_k^2}{(1-\gamma)^4}\|Q_k - Q^k\| + \frac{\eta_k}{1-\gamma}\langle Q_k - Q^k, \nabla Q^k(\mathbf{1}_k \odot A_k)\rangle \Big], \quad \text{(from Algorithm 1)} \\
&\leq E\Big[ \frac{2L_2^q\eta_k^2}{(1-\gamma)^4}\|Q_k - Q^k\| + \frac{\eta_k}{1-\gamma}\langle Q_k - Q^k, \nabla Q^k(d^k \odot A_k)\rangle \Big], \quad \text{(Conditional expectation, } (s_k, a_k) \sim d^k \text{ )} \\
&\leq E\Big[ \frac{2L_2^q\eta_k^2}{(1-\gamma)^4}\|Q_k - Q^k\| + \frac{\eta_k}{1-\gamma}\|Q_k - Q^k\|\|\nabla Q^k(d^k \odot A_k)\| \Big], \quad \text{(Cauchy Schwartz)} \\
&\leq \frac{2L_2^q\eta_k^2}{(1-\gamma)^4}\sqrt{E\|Q_k - Q^k\|^2} + \frac{\eta_k}{1-\gamma}\sqrt{E\|Q_k - Q^k\|^2}\sqrt{E\|\nabla Q^k(d^k \odot A_k)\|^2}, \quad \text{(Jensen and Cauchy inequalities )} \\
&\leq \frac{2L_2^q\eta_k^2}{(1-\gamma)^4}\sqrt{E\|Q_k - Q^k\|^2} + \frac{2\gamma\sqrt{S}A\eta_k}{(1-\gamma)^3}\sqrt{E\|Q_k - Q^k\|^2}\sqrt{E\|\nabla J^k\|^2}, \quad \text{(using Proposition B.13 )}
\end{aligned}
$$

To summarize, we have the following recursion:

$$
z_{k+1}^2 \leq (1 - 2\lambda\beta_k)z_k^2 + 2c_u^2\beta_k^2 + 2c_q^2\eta_k^2 + \frac{2L_2^q}{(1-\gamma)^4}\eta_k^2 z_k + \frac{2\gamma\sqrt{S}A}{(1-\gamma)^3}\eta_k y_k z_k.
$$

$\square$

**Proposition B.13.**

$$
\|\nabla Q^k(d^k \odot A_k)\|^2 \leq \frac{4\gamma^2 SA^2}{(1-\gamma)^4}\|\nabla J^k\|^2.
$$

*Proof.* From definition, we have

$$
Q^\pi(s, a) = R(s, a) + \gamma\sum_{s'}P(s'|s, a)v^\pi(s') \tag{50}
$$

$$
\implies \frac{d}{d\theta(s'', a'')}Q^\pi(s, a) = \gamma\sum_{s'}P(s'|s, a)\frac{d}{d\theta(s'', a'')}v^\pi(s') \tag{51}
$$

$$
= \frac{\gamma}{1-\gamma}\sum_{s'}P(s'|s, a)d_{s'}^\pi(s'')A^\pi(s'', a''). \tag{52}
$$

This implies that

$$\|\nabla Q^k (d^k \odot A_k)\|^2 = \sum_{s,a} \Big( \sum_{s'',a''} \frac{dQ^k(s,a)}{d\theta(s'',a'')} d^k(s'',a'') A_k(s'',a'') \Big)^2 \tag{53}$$

$$= \frac{1}{(1-\gamma)^2} \sum_{s,a} \Big( \sum_{s'',a''} \gamma \sum_{s'} P(s'|s,a) d^k_{s'}(s'') A^k(s'',a'') d^k(s'',a'') A_k(s'',a'') \Big)^2, \qquad \text{(putting back the value )} \tag{54}$$

$$\le \frac{\gamma^2}{(1-\gamma)^2} \sum_{s,a} \Big( \sum_{s'',a''} \sum_{s'} P(s'|s,a) d^k_{s'}(s'') d^k(s'',a'') |A^k(s'',a'')| |A_k(s'',a'')| \Big)^2, \qquad \text{(taking absolute values)} \tag{55}$$

$$= \frac{4\gamma^2 SA}{(1-\gamma)^4} \Big( \sum_{s'',a'',s'} P(s'|s,a) d^k_{s'}(s'') d^k(s'',a'') |A^k(s'',a'')| \Big)^2 \tag{56}$$

$$\le \frac{4\gamma^2 SA^2}{(1-\gamma)^4} \sum_{s'',a'',s'} P(s'|s,a) d^k_{s'}(s'') \Big( d^k(s'',a'') A^k(s'',a'') \Big)^2, \tag{57}$$

$$\text{(from Jensen, as } \sum_{s'',a'',s'} P(s'|s,a) d^k_{s'}(s'') = A) \tag{58}$$

$$\le \frac{4\gamma^2 SA^2}{(1-\gamma)^4} \sum_{s'',a''} \Big( d^k(s'',a'') A^k(s'',a'') \Big)^2, \qquad ( \text{ as } P(s'|s,a) d^k_{s'}(s'') \le 1) \tag{59}$$

$$= \frac{4\gamma^2 SA^2}{(1-\gamma)^4} \|\nabla J^k\|^2. \tag{60}$$

$\square$

## C  Solving Recursions

In this section, we solve the recursions derived above.

### C.1  Proof of Lemma 4.4

**Lemma C.1.** *The following recursions*

$$a_{k+1} \le a_k - \frac{\eta_k}{1-\gamma} y_k^2 + \frac{2}{1-\gamma} \eta_k y_k z_k + \frac{4L\eta_k^2}{(1-\gamma)^4}$$

$$a_k \le c_g y_k$$

$$z_{k+1}^2 \le (1 - 2\lambda\beta_k) z_k^2 + 2c_u^2 \beta_k^2 + 2c_q^2 \eta_k^2 + \frac{2L_2^q}{(1-\gamma)^4} \eta_k^2 z_k + \frac{2\gamma\sqrt{S}A}{(1-\gamma)^3} \eta_k y_k z_k,$$

*implies*

$$a_k \le c_l^{-\frac{2}{3}} \Big( \max\{c_\eta, 2c_l^2 u_0^3\} \Big)^{\frac{1}{3}} \Big( \frac{1}{\frac{1}{\alpha_0^3} + 2k} \Big)^{\frac{1}{3}} = \Big( \max\{c_l^{-\frac{2}{3}} c_\eta^{\frac{1}{3}}, 2u_0\} \Big) \Big( \frac{1}{\frac{1}{\alpha_0^3} + 2k} \Big)^{\frac{1}{3}},$$

*with constants $\alpha_0, c_l, c_2$ defined in Table 3.*

*Proof.* Adding the first and last recursions, and using $z_k \leq c_z$ from Table 3, we get

$$a_{k+1} + z_{k+1}^2$$

$$\leq a_k + z_k^2 - \frac{\eta_k y_k^2}{1-\gamma} - 2\lambda\beta_k z_k^2 + 2c_u^2\beta_k^2 + \left( \frac{4L}{(1-\gamma)^4} + 2c_q^2 + \frac{2L_2^q c_z}{(1-\gamma)^4} \right) \eta_k^2 + \left( \frac{2\gamma\sqrt{S}A}{(1-\gamma)^3} + \frac{2}{1-\gamma} \right) \eta_k y_k z_k$$

$$\leq a_k + z_k^2 - \frac{\eta_k y_k^2}{1-\gamma} - 2\lambda\beta_k z_k^2 + 2c_u^2\beta_k^2 + \left( \frac{4L}{(1-\gamma)^4} + 2c_q^2 + \frac{2L_2^q c_z}{(1-\gamma)^4} \right) \eta_k^2 + \frac{3\sqrt{S}A}{(1-\gamma)^3}\eta_k y_k z_k, \qquad \text{(simplifying)}$$

$$\leq a_k + z_k^2 - \eta_k \left[ \frac{y_k^2}{1-\gamma} + 2\lambda c_\beta z_k^2 - \frac{3\sqrt{S}A}{(1-\gamma)^3}y_k z_k \right]$$

$$+ \left( \underbrace{2c_u^2 c_\beta^2 + \frac{4L}{(1-\gamma)^4} + 2c_q^2 + \frac{2L_2^q c_z}{(1-\gamma)^4}}_{:=c_\eta} \right) \eta_k^2, \qquad \text{(as } \frac{\beta_k}{\eta_k} = c_\beta)$$

$$\leq a_k + z_k^2 - \frac{\eta_k}{2} \left[ \frac{y_k^2}{(1-\gamma)} + 2\lambda c_\beta z_k^2 \right] - \frac{\eta_k}{2} \left[ \frac{y_k^2}{(1-\gamma)} + 2\lambda c_\beta z_k^2 - \frac{6\sqrt{S}A}{(1-\gamma)^3}y_k z_k \right] + c_\eta\eta_k^2, \qquad \text{(plus-minus)}$$

$$\leq a_k + z_k^2 - \frac{\eta_k}{2} \left[ \frac{y_k^2}{(1-\gamma)} + 2\lambda c_\beta z_k^2 \right] - \eta_k \left[ \underbrace{\sqrt{\frac{2\lambda c_\beta}{(1-\gamma)}} - \frac{3\sqrt{S}A}{(1-\gamma)^3}}_{\geq 0 \text{ as } c_\beta := \frac{9SA^2}{2(1-\gamma)^5}} \right] y_k z_k + c_\eta\eta_k^2, \qquad \text{(as } a + b \geq 2\sqrt{ab})$$

$$= a_k + z_k^2 - \frac{\eta_k}{2} \left[ \frac{y_k^2}{(1-\gamma)} + 2\lambda c_\beta c_z^2 (\frac{z_k}{c_z})^2 \right] + c_\eta\eta_k^2, \qquad \text{(divide-multiply)}$$

$$= a_k + z_k^2 - \frac{\eta_k}{2} \left[ \frac{y_k^2}{(1-\gamma)} + 2\lambda c_\beta c_z^2 (\frac{z_k}{c_z})^4 \right] + c_\eta\eta_k^2, \qquad \text{(as } \frac{z_k}{c_z} \leq 1 \text{ by defn of } c_z, \text{ see Table 3)}$$

$$\leq a_k + z_k^2 - \frac{\eta_k}{2} \left[ \frac{a_k^2}{c_g^2(1-\gamma)} + \frac{2\lambda c_\beta}{c_z^2} z_k^4 \right] + c_\eta\eta_k^2, \qquad \text{(using } a_k \leq c_g y_k)$$

$$\leq a_k + z_k^2 - 2\eta_k c_l \left[ a_k^2 + z_k^4 \right] + c_\eta\eta_k^2, \qquad \text{(as } c_l := \frac{1}{4}\min\{\frac{1}{c_g^2(1-\gamma)}, \frac{2\lambda c_\beta}{c_z^2}\})$$

$$\leq a_k + z_k^2 - c_l\eta_k \left( a_k + z_k^2 \right)^2 + c_\eta\eta_k^2, \qquad \text{(using } (a+b)^2 \leq 2(a^2 + b^2)).$$

Taking $u_k = a_k + z_k^2, \omega_k = \sqrt{\eta_k}$, the above recursion is of the form:

$$u_k \leq u_k - c_l\eta_k u_k^2 + \frac{1}{2}c_\eta\eta_k^2. \tag{61}$$

Taking $c_1 = c_l$ and $c_2 = \max\{c_\eta, 2c_l^2 u_0^3\}$ to ensure $\alpha_0 = c_1^{\frac{2}{3}}c_2^{-\frac{1}{3}}u_0 \leq 2^{-\frac{1}{3}}$ in Lemma C.2, we get

$$u_k \leq c_l^{-\frac{2}{3}} \left( \max\{c_\eta, 2c_l^2 u_0^3\} \right)^{\frac{1}{3}} \left( \frac{1}{\frac{1}{\alpha_0^3} + 2k} \right)^{\frac{1}{3}}. \tag{62}$$

Note that $a_k \leq u_k$ as $z_k^2 \geq 0$, yielding the desired result. $\qquad \square$

**Lemma C.2.** *[ODE domination for Recursion] Given* $\frac{d\alpha_x}{dx} = -\frac{1}{2}\alpha_x^4, \alpha_k = \left( \frac{1}{\frac{1}{\alpha_0^3} + 2k} \right)^{\frac{1}{3}}$, *and*
$\eta_k = c_1^{-\frac{1}{3}}c_2^{-\frac{1}{3}}\alpha_k^2$ *the recursion,*

$$u_{k+1} \leq u_k - c_1\eta_k u_k^2 + \frac{1}{2}c_2\eta_k^2,$$

*then* $u_k \leq c_1^{-\frac{2}{3}}c_2^{\frac{1}{3}}\alpha_k$ *for all* $k \geq 0$, *where* $\alpha_0 = c_1^{\frac{2}{3}}c_2^{-\frac{1}{3}}u_0 \leq 2^{-\frac{1}{3}}$

*Proof.* Let $\nu_k = c_1^{\frac{2}{3}} c_2^{-\frac{1}{3}} u_k$ and $\alpha_k = c_1^{\frac{1}{6}} c_2^{\frac{1}{6}} \sqrt{\eta_k}$. Then, multiplying both sides with $c_1^{\frac{2}{3}} c_2^{-\frac{1}{3}}$, we get

$$c_1^{\frac{2}{3}} c_2^{-\frac{1}{3}} u_{k+1} \le c_1^{\frac{2}{3}} c_2^{-\frac{1}{3}} u_k - c_1^{\frac{1}{3}} c_2^{\frac{1}{3}} \eta_k \left( c_1^{\frac{2}{3}} c_2^{-\frac{1}{3}} u_k \right)^2 + \frac{1}{2} c_1^{\frac{2}{3}} c_2^{\frac{2}{3}} \eta_k^2 \tag{63}$$

$$\implies \nu_{k+1} \le \nu_k - \alpha_k^2 \nu_k^2 + \frac{1}{2} \alpha_k^4. \tag{64}$$

Now let $f_k(\nu) = \nu - \alpha_k^2 \nu^2$ and assume, $\nu_k \le \alpha_k$, then

$$\nu_{k+1} \le f_k(\nu_k) + \frac{1}{2} \alpha_k^4 \tag{65}$$

$$\le f_k(\alpha_k) + \frac{1}{2} \alpha_k^4, \qquad (\text{as } f_k(\nu) \text{ is increasing for } \nu \le \frac{1}{2\alpha_k^2}, \text{ and } \nu_k \le \alpha_k \le \frac{1}{2\alpha_0^2} \le \frac{1}{2\alpha_k^2}) \tag{66}$$

$$= \alpha_k - \frac{1}{2} \alpha_k^4, \qquad (\text{putting the value back of } f) \tag{67}$$

$$\le \alpha_k - \int_{x=k}^{k+1} \frac{1}{2} \alpha_k^4 dx, \qquad (\text{dummy integral}) \tag{68}$$

$$= \alpha_k - \int_{x=k}^{k+1} \frac{1}{2} \alpha_x^4 dx, \qquad (\text{as } \alpha_x \text{ is decreasing}) \tag{69}$$

$$\le \alpha_k - \int_{x=k}^{k+1} \frac{1}{2} \alpha_k^4 dx, \qquad (\text{dummy integral}) \tag{70}$$

$$= \alpha_k + \int_{x=k}^{k+1} \frac{d\alpha_x}{dx} dx, \qquad (\text{as } \frac{d\alpha_x}{dx} = -\frac{1}{2} \alpha_x^4) \tag{71}$$

$$\le \alpha_{k+1}, \qquad (\text{basic calculus}). \tag{72}$$

From induction arguments, we get $\nu_k \le \alpha_k$ for all $k \ge 0$ given the base condition $\nu_0 \le \alpha_0$ is satisfied. In other words,

$$c_1^{\frac{2}{3}} c_2^{-\frac{1}{3}} u_k \le \alpha_k = \left( \frac{1}{\frac{1}{\nu_0^3} + 2k} \right)^{\frac{1}{3}}. \tag{73}$$

$\square$

## C.2 Proof of main theorem

**Theorem C.3** (Main Result). *For step size $\eta_k = O(k^{-\frac{2}{3}})$ and $\beta_k = c_\beta \eta_k$ in Algorithm 1, we have*

$$J^* - E J^{\pi_{\theta_k}} \le \max \left\{ \frac{S^{\frac{4}{3}} A^{\frac{4}{3}} C_{PL}^{\frac{4}{3}}}{c^{\frac{4}{3}} (1-\gamma)^{\frac{10}{3}}}, \frac{A^{\frac{4}{3}}}{\lambda^{\frac{2}{3}} (1-\gamma)^{\frac{6}{3}}} \right\} \frac{1}{k^{\frac{1}{3}}}, \qquad \forall k > 0.$$

*where $C$ is some numerical constant.*

*Proof.* From Lemma C.1, we have

$$J^* - E J^{\pi_{\theta_k}} = a_k \le c_l^{-\frac{2}{3}} \left( \max\{c_\eta, 2c_l^2 u_0^3\} \right)^{\frac{1}{3}} \left( \frac{1}{\frac{1}{\alpha_0^3} + 2k} \right)^{\frac{1}{3}}$$

$$\le C_1 \frac{\max\{c_l^{-\frac{2}{3}} c_\eta^{\frac{1}{3}}, u_0\}}{k^{\frac{1}{3}}}. \qquad (\text{re-arranging, } \alpha_0 > 0, C_1 \text{ is numerical constant}).$$

Putting the values from Table 3, we have

$$c_\eta^{\frac{1}{3}} \le \frac{10 S^{\frac{2}{3}} A^{\frac{4}{3}}}{(1-\gamma)^4},$$

and

$$c_l^{-\frac{2}{3}} = \left[ \frac{1}{4(1-\gamma)} \min\{\frac{c^2}{SC_{PL}^2}, \frac{9\lambda S}{4(1-\gamma)^2}\} \right]^{-\frac{2}{3}} \tag{74}$$

$$\leq 2^{\frac{4}{3}} (1-\gamma)^{\frac{2}{3}} \max\{\frac{S^{\frac{2}{3}} C_{PL}^{\frac{4}{3}}}{c^{\frac{4}{3}}}, \frac{2^{\frac{4}{3}}(1-\gamma)^{\frac{4}{3}}}{3^{\frac{4}{3}}\lambda^{\frac{2}{3}}S^{\frac{2}{3}}}\}. \tag{75}$$

Hence, we get

$$c_l^{-\frac{2}{3}} c_\eta^{\frac{1}{3}} \leq C_1 \frac{S^{\frac{2}{3}} A^{\frac{4}{3}}}{(1-\gamma)^{\frac{10}{3}}} \left[ \max\{\frac{S^{\frac{2}{3}} C_{PL}^{\frac{4}{3}}}{c^{\frac{4}{3}}}, \frac{(1-\gamma)^{\frac{4}{3}}}{\lambda^{\frac{2}{3}}S^{\frac{2}{3}}}\} \right] \tag{76}$$

$$\leq C_1 \max\left\{ \frac{S^{\frac{4}{3}} A^{\frac{4}{3}} C_{PL}^{\frac{4}{3}}}{c^{\frac{4}{3}}(1-\gamma)^{\frac{10}{3}}}, \frac{A^{\frac{4}{3}}}{\lambda^{\frac{2}{3}}(1-\gamma)^{\frac{6}{3}}} \right\}. \tag{77}$$

and $u_0 = O(\frac{SA}{(1-\gamma)^2})$, hence the complexity is

$$a_k \leq C_1 \max\left\{ \frac{S^{\frac{4}{3}} A^{\frac{4}{3}} C_{PL}^{\frac{4}{3}}}{c^{\frac{4}{3}}(1-\gamma)^{\frac{10}{3}}}, \frac{A^{\frac{4}{3}}}{\lambda^{\frac{2}{3}}(1-\gamma)^{\frac{6}{3}}} \right\} \frac{1}{k^{\frac{1}{3}}},$$

where $C_2$ is some numerical constant. For comparision, the iteration complexity for the exact gradient case is $O(\frac{SC_{PL}^2}{c^2(1-\gamma)^6 k})$ as shown in Theorem 4 of Mei et al. (2020). □

Notably, actor-critic dependence little better in mis-match coefficient $C_{PL}$ (yes, we double checked), and only slightly expensive in state space and horizon.

## D  Numerical Simulations

This section numerically illustrate with convergence rate of single-time-scale Algorithm 1 with different step size schedule. All MDPs have randomly generated transition kernel and reward function, with codes available at `https://anonymous.4open.science/r/AC-C43E/`. For simplicity, the samples are generated uniformly instead of discounted occupation measure.

Figure 2 illustrates that the learning rate $\eta_k = \beta_k = k^{-\frac{2}{3}}$ has the best performance. Notably, slow decaying learning rates such as $\eta_k = \beta_k = 0.01k^0, k^{-\frac{1}{3}}, k^{-\frac{1}{2}}$ have better performance in the starting, and eventually they surpassed by $\eta_k = \beta_k = k^{-\frac{2}{3}}$. In addition, $\eta_k = \beta_k = k^{-1}$ has the worst performance.

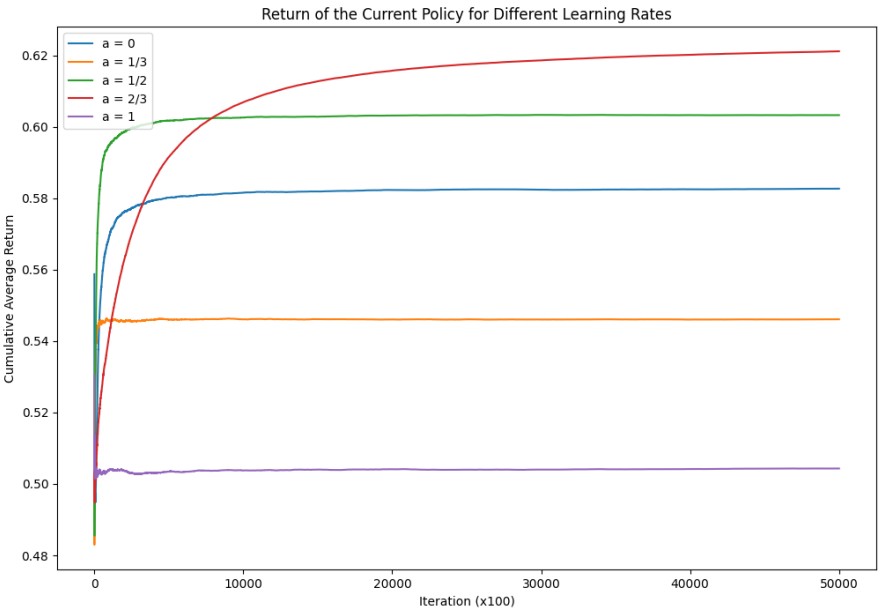

Figure 2: Convergence Rate of Algorithm 1, on random MDP with state space =50, action space = 5, learning rate $\eta_k = \beta_k = k^{-a}$

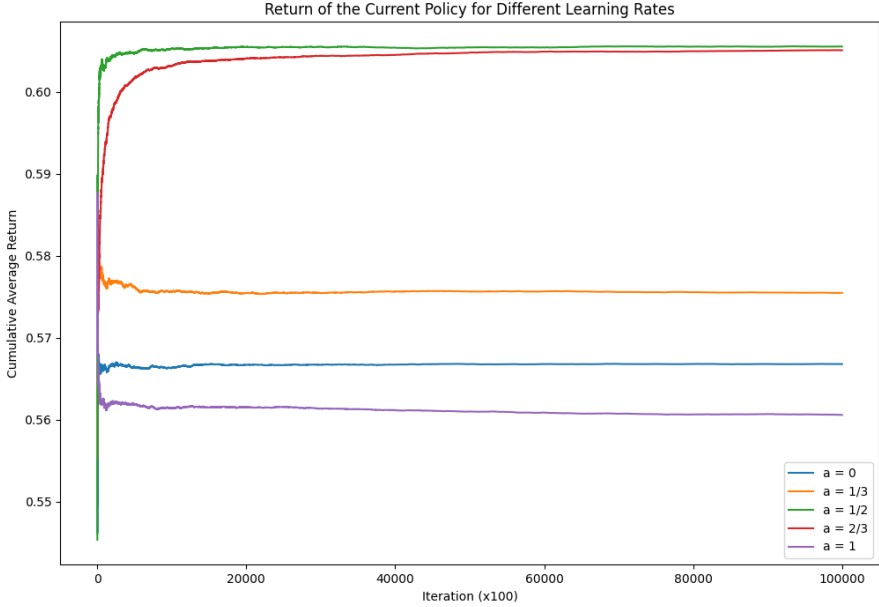

Figure 3: Convergence Rate of Algorithm 1, on random MDP with state space =5, action space = 2, learning rate $10\eta_k = \beta_k = k^{-a}$

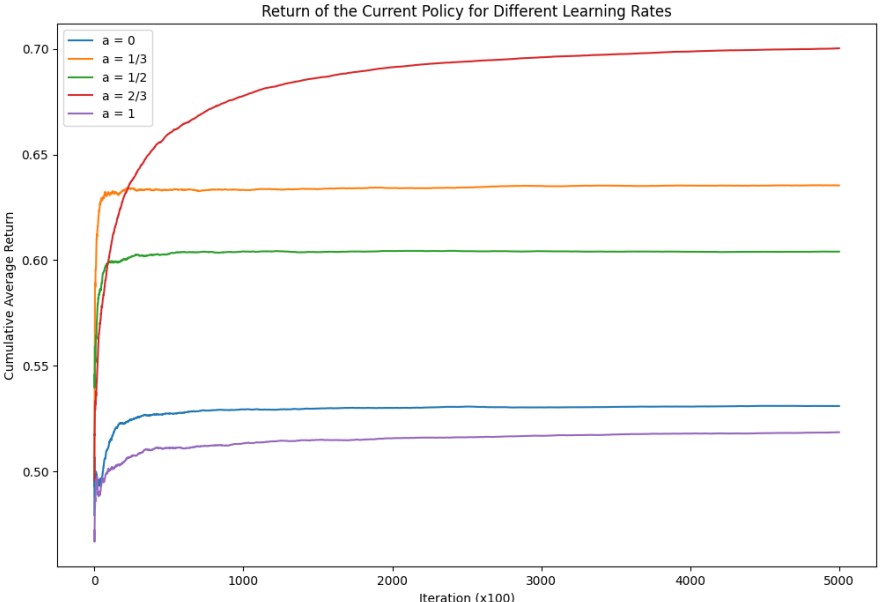

Figure 4: Convergence Rate of Algorithm 1, on random MDP with state space =20, action space = 5, learning rate $\eta_k = \beta_k = k^{-a}$.

