# OpenReview forum: "On the Convergence of Single-Timescale Actor-Critic"
_NeurIPS.cc/2025/Conference — NeurIPS 2025 poster_

### Official Review · Reviewer_ZXNM · 2025-06-08

**Clarity:** 1
**Significance:** 4
**Originality:** 3
**Rating:** 4
**Confidence:** 4

**Summary:**

The paper studies the standard actor-critic algorithm with the step sizes for the policy updates and value function updates on the same timescale, and shows that the algorithm's last iterate converges with rate $\mathcal{O}(1/k^{1/3})$ (measured in the function value space). This is the best complexity derived so far in the literature.

**Questions:**

1) Can the authors clarify if I correctly understand the innovation of the paper in my comments above (in Strengths & Weaknesses)? Specifically, the limitation in prior works like Chen and Zhao [2024] is that they disregard the gradient domination structure and treat the policy optimization objective as a standard smooth non-convex function, but the current paper leverages the structure in the analysis of the actor. From this perspective, the key novelty of the paper lies in Lemma 4.1. To my knowledge, Lemma 4.2 is a standard result (namely, showing the approximate iteration-wise critic convergence up to gradient norm of the actor). The fact that Lemma 4.2 is standard does not undermine the contribution of the paper. For clarity, I strongly encourage the authors to be precise and transparent in communicating to the audience which parts of the analysis are novel and which rely on established results. If this is indeed the case, the authors should consider substantially revising the paper to place more emphasis on Lemma 4.1 and its analysis in the main paper.

2) The authors state "Additionally, when this ODE tracking method is applied to the recursion for the exact gradient case, it produces better results compared to existing bounds, such as those in Mei et al. (2022)." in line 52-54, but never establish anything along this line.

3) In line 119-121, the authors make the comment "Yuan et al. (2022) too establishes global convergence with sample complexity of $\mathcal{O}(\epsilon^{-3})$, however, it requires an additional structural assumption on the problem which is highly restrictive." The sentence is written in a way that makes the audience want to know the exact assumption. Please provide more detail here.

4) In line 150-151, "However, the return ... leading to the following result." -- the gradient domination lemma is not a direct consequence of $J$ being smooth.

5) I suggest that the authors re-organize the appendix and add proper linkage between lemmas/propositions and their proofs, to make it easier to the audience to locate the proofs.

6) Something nice about the result in this paper is that the convergence is established on the last iterate. If we follow the analysis in Chen and Zhao [2024] and plugs in the gradient domination condition at the end, the convergence in value function space will be on the best iterate (in addition to having an inferior rate). The authors should consider mentioning this additional advantage of their analysis.

**Ethical Concerns:**

["NO or VERY MINOR ethics concerns only"]

**Final Justification:**

Raising score as the authors' response provides more clarity on the technical innovation

**Limitations:**

I do not foresee any negative societal impact.

**Paper Formatting Concerns:**

No formatting concerns.

**Quality:**

2

**Strengths And Weaknesses:**

The strength lies in the result itself. The convergence rate $\mathcal{O}(1/k^{1/2})$ measured by gradient norm squared has been the best-known result and this paper establishes $\mathcal{O}(1/k^{1/3})$ convergence in function value space, which is highly significant. I believe many researchers in the RL theory will find the result and analysis interesting.

On the flip side, the paper is badly written, not in terms of the language, but in delivering a clear message on what specific technique/analysis enables the improved rate. The authors The authors repeatedly emphasize the challenges arising from coupled actor and critic variables and from using single-loop single-timescale updates. I do not believe these are the actual challenges. As the authors themselves noted, Chen and Zhao [2024] analyzes the same single-loop single-timescale algorithm and establishes convergence. Moreover, in broader settings of stochastic approximation/optimization, under structures like non-convexity with Lipschitz gradients, PL condition, and strong convexity, many other papers consider a similar single-timescale algorithm and show that the algorithm achieves the optimal convergence rates.

The weakness in the analysis from Chen and Zhao [2024] is that they treat the policy optimization objective as a general smooth non-convex function without using the gradient domination structure (or one may plug in the inequality at the end). In my understanding, the core innovation of the current paper is to exploit this structure when bounding the iteration-wise drift of the actor. A more thorough discussion of this aspect should be the actual focus of the presentation.

Overall, I believe that the results derived in the paper are very much worth being known to the community, but significant additional work is needed to improve the presentation and to help the audience understand the core contribution and technical innovation.

References

[1] Shen, H. and Chen, T., 2022. A single-timescale analysis for stochastic approximation with multiple coupled sequences. Advances in Neural Information Processing Systems, 35, pp.17415-17429.

[2] Zeng, S. and Doan, T., 2024, June. Fast two-time-scale stochastic gradient method with applications in reinforcement learning. In The Thirty Seventh Annual Conference on Learning Theory (pp. 5166-5212). PMLR.

[3] Huang, Y., Wu, Z., Ma, S. and Ling, Q., 2024. Single-Timescale Multi-Sequence Stochastic Approximation Without Fixed Point Smoothness: Theories and Applications. arXiv e-prints, pp.arXiv-2410.

---

> ### Author Rebuttal · Authors · 2025-07-30
>
> We sincerely thank the reviewer for their thorough and insightful feedback. We are encouraged that they find our results and analysis both significant and of interest to the reinforcement learning theory community. We also greatly appreciate the constructive criticism and thoughtful suggestions, which will help us improve the final version of the paper.
>
> ***Q1) The weakness in the analysis from Chen and Zhao [2024] is that they treat the policy optimization objective as a general smooth non-convex function without using the gradient domination structure (or one may plug in the inequality at the end). In my understanding, the core innovation of the current paper is to exploit this structure when bounding the iteration-wise drift of the actor. A more thorough discussion of this aspect should be the actual focus of the presentation.***
> > We thank the reviewer for this insightful observation. The comment is indeed correct in noting that our analysis leverages the Gradient Domination Lemma (GDL) structure from the outset, using it as a foundation to derive and solve the core recursion. We will emphasize this point more clearly in the final version.
>
> > This approach stands in contrast to that of Chen and Zhao [2024], who rely on standard smooth optimization techniques and employ a telescoping sum to upper bound the gradient norm. In their analysis, the GDL is applied only at the end, yielding a best-iterate guarantee but leading to a weaker overall convergence rate.
>
> > We will incorporate this comparison into the introduction, main text, and discussion sections of the final version.
>
> ***Q2) Can the authors clarify if I correctly understand the innovation of the paper in my comments above (in Strengths & Weaknesses)? Specifically, the limitation in prior works like Chen and Zhao [2024] .... more emphasis on Lemma 4.1 and its analysis in the main paper.***
> > Yes, the reviewer is correct in understanding both the novelty and the techniques used in the paper. Indeed, Lemma 4.2 is derived using standard techniques that are well-established in the literature. We appreciate the suggestion to emphasize Lemma 4.1 more prominently, and we will incorporate this in the final version.
>
> ***Q3) The authors state "Additionally, when this ODE tracking method is applied to the recursion for the exact gradient case, it produces better results compared to existing bounds, such as those in Mei et al. (2022)." in line 52-54, but never establish anything along this line.***
> > We apologize for this omission. By applying our techniques, we can improve the iteration complexity by a factor of $\frac{1}{(1 - \gamma)^3}$ compared to Mei et al. (2022), along with better dependence on the initial sub-optimality. A brief explanation follows:
>
> >> In the exact gradient case, from the sufficient increase lemma and using a step size $\eta_k = \frac{1}{L}$, where $L = \frac{8}{(1 - \gamma)^3}$ is the smoothness coefficient, we have:
> $$ a_{k+1} - a_k \geq \frac{\lVert \nabla J^{\pi_{\theta_k}} \rVert_2^2}{2L}. $$
> From the Gradient Domination Lemma (GDL), we also have: $$ a_k \leq \frac{\sqrt{S} C_{PL}}{c} \lVert \nabla J^{\pi_{\theta_k}} \rVert_2.
>  $$ Combining the two gives the recursion: $$
>  a_{k+1} - a_k \geq \frac{c^2}{2 L S C_{PL}^2} a_k^2. $$
>  Using ODE tracking, it follows that:
>  $$ a_k \leq \frac{1}{\frac{1}{a_0} + \frac{c^2}{2 L S C_{PL}^2} k}
>  = \frac{1}{\frac{1}{a_0} + \frac{c^2 (1 - \gamma)^3}{16 S C_{PL}^2} k}, \qquad \forall k \geq 0.$$
>
> > In comparison, Mei et al. (2022) report:$$
>  a_k \leq \frac{16 S C_{PL}^2}{c^2 (1 - \gamma)^6 k}. $$
>  Our result offers improvements in two important aspects:
> 1. **Improved horizon dependence:**
>      Our bound improves the dependence on the discount factor from $(1 - \gamma)^{-6}$ to $(1 - \gamma)^{-3}$:
>      $$a_k \leq \frac{16 S C_{PL}^2}{c^2 (1 - \gamma)^3 k}.$$
>  2. **Tighter behavior in early iterations:**
>      The bound from Mei et al. (2022),
>      $$\frac{16 S C_{PL}^2}{c^2 (1 - \gamma)^6 k}, $$
>      can be larger than the maximum possible sub-optimality (which is upper bounded by $\frac{2}{1 - \gamma}$), especially when the initial iterate is far from optimal. This results in a vacuous bound for early iterations. In contrast, our result yields a meaningful and tighter upper bound starting from the very first iteration.
>
> > We have incorporated this discussion, along with the corresponding ODE tracking lemma and its proof, into the appendix of the final version.
>
> ***Q4) In line 119-121, the authors make the comment "Yuan et al. (2022) too establishes global convergence with sample complexity of $\mathcal{O}(\epsilon^{-3})$, however, it requires an additional structural assumption on the problem which is highly restrictive." The sentence is written in a way that makes the audience want to know the exact assumption. Please provide more detail here.***
> > The results in Yuan et al. (2022) consider stochastic policy gradient descent under various assumptions, using a batch size $m$ and horizon length $H$. Their analysis applies to a setting with only an actor parameter (i.e., no critic).
>
> > **Corollary 4.11** (under ABC Assumption 3.3):
> > The most basic global convergence result yields a sample complexity of   $ O(\epsilon^{-6})$   with horizon length $H = \frac{\log(\epsilon^{-1})}{1 - \gamma}$.
>
> > **Corollary 3.7** (under ABC Assumption 3.3 and Assumption 3.6 with slackness $\epsilon'$):
> > - *Limiting case:* When $\epsilon' = 0$, corresponding to the infinite-horizon case, the policy converges to within $O(\epsilon)$ sub-optimality in $O(\epsilon^{-3})$ iterations. However, this requires infinite-length trajectories at each iteration, making the total sample complexity essentially unbounded.
> > - *General case:* When $\epsilon' > 0$, the policy converges to within $O(\epsilon + \epsilon')$ sub-optimality in   $ O(\epsilon^{-1} (\epsilon')^{-2})$   iterations. This means the algorithm cannot be guaranteed to converge to arbitrarily small sub-optimality unless $\epsilon' \to 0$ (i.e., infinite-horizon setting).
>
> > In summary, the stochastic policy gradient algorithm in Yuan et al. (2022) is substantially different from ours, which focuses on single time-scale actor-critic methods. Our results either achieve better sample complexity or require fewer/less restrictive assumptions. We will clarify this comparison more precisely in the final version.
>
> ***Q5) In line 150-151, "However, the return ... leading to the following result." -- the gradient domination lemma is not a direct consequence of $J$ being smooth.***
> > Thank you for pointing this out. This line is indeed misplaced, and we will remove it in the updated version.
>
> ***Q6) I suggest that the authors re-organize the appendix and add proper linkage between lemmas/propositions and their proofs, to make it easier to the audience to locate the proofs.***
> > Thank you very much for the helpful suggestion. We will include an appendix outline and clearly indicate the correspondence between technical results in the appendix and their counterparts in the main text in the updated version.
>
> ***Q7)Something nice about the result in this paper is that the convergence is established on the last iterate.... The authors should consider mentioning this additional advantage of their analysis.***
> > Thank you for the suggestion. We will include this point following our main result, Theorem 3.3, in the updated version.

---

> > ### Comment · Reviewer_ZXNM · 2025-08-01
> >
> > I appreciate the authors' response, especially the clarification on Q3. However, the responses to Q1 and Q2 remain insufficient and do not fully address the concern. Even though I understand Lemma 4.1 is the contribution, I do not understand how technical innovations are made and what the technical innovations exactly are. The authors should consider providing a sketch of how Lemma 4.1 improves the actor convergence analysis.

---

> ### Author Response · Authors · 2025-08-01
> **Technical Innovation of the paper.**
>
> We thank the reviewer for the prompt response. We are glad to know that the reviewer is satisfied with the answers to most questions, except Q1 and Q2, which we now address below.
>
> ---
>
>  **Novelty of the Paper**
>
> First, we apologize for any confusion caused by our previous response. To clarify: **Lemmas 4.1, 4.2, and 4.3 are supporting results** that help derive the interdependent recursions summarized in Equation (3). The **main technical novelty** lies in our method for solving these coupled recursions, culminating in **Lemma 4.4**.
>
>
>
> > **Creative Step**
> In prior work, notably by Chen and Zhao [2024], the analysis leverages the *possibility of a telescoping sum* in the actor update recursion ($a_{k+1} - a_k$). In contrast, our approach introduces a novel **Lyapunov function** of the form $a_k + z_k^2$, which leads to a significantly more elegant and well-behaved recursion.
> Identifying this Lyapunov structure was far from straightforward—it required substantial insight and effort. Once identified, it allowed us to bypass the major bottleneck in solving the coupled recursions.
>
>
> > **Why Solving Equation (3) Is Difficult Without the Lyapunov Term**: Equation (3) consists of coupled recursions for the actor and critic. To bound the actor term ($a_k$), we require a **lower bound on $y_k$**, which we obtain via the GDL (Lemma 4.3). However, to control the critic error ($z_k^2$), the critic recursion requires an **upper bound on $y_k$**, which is **not available**.
> This introduces a major hurdle in analyzing the recursion. By summing the actor and critic recursions and carefully constructing the Lyapunov term $a_k + z_k^2$, we eliminate $y_k$ entirely from the recursion without its upper bound—achieving a breakthrough that would not be possible otherwise (see Line 279).
>
> > **Technical Contribution**: Once the Lyapunov recursion is derived (Line 280), solving it remains challenging due to the **time-varying learning rate** $\eta_k$. While classical intuition suggests that such a difference equation should follow its corresponding ODE under small step sizes, **Lemma 4.4 formalizes this intuition in a rigorous and general way**.
> The **ODE tracking lemma (Lemma 4.4)** is a core technical contribution of this work. Both the **result itself** and the **proof technique** used to derive it are, in our view, elegant and novel.
>
> ---
>
> ### **Supporting Results**
>
> > **Lemma 4.1 (Actor Recursion):**  Derived from the smoothness of the return function, this lemma ensures monotonic improvement under sufficiently small step sizes. It is a noisy-biased variant of the standard *sufficient increase lemma*, and similar results have appeared in various forms in the literature.
>
> > **Lemma 4.2 (Critic Recursion):**   This follows standard techniques for convergence of the critic using sample-based updates. It relies on the contraction property of the Bellman operator for policy evaluation.
>
> > **Lemma 4.3 (GDL Recursion):**   This lemma applies a gradient domination argument in expectation, using standard tools such as Jensen’s inequality.
>
> ---
>
> ### **Summary**
>
> To avoid confusion, we are considering **relabeling Lemmas 4.1, 4.2, and 4.3 as propositions** and making it explicit that these results are derived using standard techniques. This would help highlight that:
>
> > The **key insight** lies in identifying the Lyapunov function that bypasses the need for an upper bound on $\|y_k\|$.
>
> > The **main technical novelty** lies in **Lemma 4.4**, which solves the Lyapunov recursion via a general ODE tracking argument.
>
> We hope this clarifies the core contributions and addresses the remaining concerns in Q1 and Q2. Thank you again for the thoughtful feedback. We are glad to answer any further questions, if any.

---

> > ### Comment · Reviewer_ZXNM · 2025-08-02
> >
> > I appreciate the authors' clarification. I would ideally like to see the revised version of the paper, but I understand the constraints as this is a conference submission. I will raise my score to a weak accept, but I strongly suggest that the authors significantly rewrite the introduction and presentation of technical results to increase the impact of the work and to make it easier for researchers in the community to digest the innovation.

---

> > > ### Author Response · Authors · 2025-08-03
> > >
> > > We are glad and thankful for the reviewer's feedback (and also for summarizing some part of the work better than us). We are looking forward to incorporating the reviewers suggestions in the final draft, particulary on presenting the technical novelty of the work.

---

### Official Review · Reviewer_ZGMW · 2025-06-11

**Clarity:** 2
**Significance:** 3
**Originality:** 3
**Rating:** 4
**Confidence:** 2

**Summary:**

The paper considers the sample-complexity for the problem of finding a $\varepsilon$-optimal policy for softmax-parameterised infinite-horizon discounted Markov decision processed. They focus on the sample-complexity’s dependence on $\varepsilon$ and establish that a single-timescale actor-critic aglorithm achieves a sample-complexity of $O(\varepsilon^{-3})$ improving upon the best known results of $O(\varepsilon^{-4})$ from prior work. This is achieved by an analysis that directly bounds the sub-optimality gap (instead of convergence to stationary points) and combines recursions for the actor and the critic simultaneously (the Lyapunov term).

**Questions:**

- See weaknesses for some suggestions on improving presentation. This paper has interesting results but suffers from the quality of the presentation. If the presentation of the paper was improved, it would have a higher score.
- In the  discussion following Lemma 4.1, it is not clear to me why the last term “accounts for the variance in the stochastic update of the policy” since in the analysis this term arises from the smoothness of the value and not from the stochasticity of the update ?
- You mention that you believe that a bias analysis may lead to the optimal $O(\varepsilon^{-2})$ sample-complexity. Do you believe this is achievable without any modification to the algorithm ? Is there perhaps a slight adaptation of the algorithm that could improve your variance analysis ?

**Ethical Concerns:**

["NO or VERY MINOR ethics concerns only"]

**Final Justification:**

The work presents an interesting and significant set of results that I believe are worthy of acceptance. My main concern is the quality of presentation, which in the current version is below the standard expected. The authors have responded that they will improve this - it is hard to assess how much it will actually be improved. I recommend acceptance conditioned on substantial work being done to improve presentation quality.

**Limitations:**

Yes

**Quality:**

2

**Strengths And Weaknesses:**

Strengths:
- The problem considered is of practical relevance: the single-timescale version of actor-critic more closely resembles algorithms used in practice than the two-timescale version.
- The sample-complexity of the single-timescale actor-critic algorithm is improved from $O(\varepsilon^{-4})$ to $O(\varepsilon^{-3})$.
- The provided analysis and key idea is novel, fairly simple, insightful and general.
- The work generally does a good job of presenting the setting and contextualising their results within prior work.

Weaknesses:
- The sample-complexity does not match the lower-bound whose dependence in $O(\varepsilon^{-2})$. The dependence on quantities other than $\varepsilon$ are ignored. In particular, the dependence on the effective horizon $1/(1-\gamma)$ and the size of the state-action space $SA$, which are hidden within constants, appears like it could be quite poor. While I understand that it could be argued that until the optimal dependence on $\varepsilon$ is achieved, the dependence on these other quantities is of secondary importance, having a discussion on the explicit dependence on these in the final bound would be helpful for the reader.
- Presentation: the writing / clarity of presentation is below the standard expected. The appendix has quite a few repeated entries (one reference even appears three identical times), some references are missing their publication venues… etc. There are many typos and grammatical errors (e.g. l.70, l.78, l.81-84, l.266… etc). Table 4 referenced at l.242 does not exist or points to the wrong thing – I guess Table 2 was meant. At the top of page 7, $Q_k$ has not been formally defined - I guess it is the update in Algo 1 but then would be clearer to write in Algo 1 with this notation. In the Convergence Analysis section, it would be useful to have pointers to where the specific sub-results are proved in the appendix. Some of the discussions are a bit unclear / hard to follow. In particular, the discussion l.268-272 is not easy to understand.

---

> ### Author Rebuttal · Authors · 2025-07-30
>
> We thank the reviewer for the encouraging reviews. We are glad that the reviewer finds key ideas and analysis to be novel, fairly simple, and insightful. We're also happy to hear that the presentation of the setting and positioning within prior work were clear.
>
>
>
> ***Q1) The sample-complexity does not match the lower-bound whose dependence in $O(\varepsilon^{-2}).$***
> > The reviewer is correct in noting that our $O(\epsilon^{-3})$ complexity does not match the lower bound of $O(\epsilon^{-2})$. However, our result significantly improves upon the existing complexity of $O(\epsilon^{-4})$, thereby narrowing the gap. To the best of our knowledge, this represents the current state-of-the-art for single-time-scale actor-critic algorithms.
>
> ***Q2) The dependence on quantities other than $\varepsilon$ are ignored. In particular, the dependence on the effective horizon $1/(1-\gamma)$ and the size of the state-action space $SA$, which are hidden within constants, appears like it could be quite poor.***
> > All prior works report complexity only in terms of $\epsilon$, and we followed the same for consistency and clarity. Nonetheless, the sample complexity for our actor-critic algorithm is:
> >
> > $$
> > O\left(\frac{S^{4/3} A^{2/3} C_{PL}^{4/3}}{\lambda^{2/3} (1 - \gamma)^{16/3} \, k^{1/3}}\right).
> > $$
> >
> > For comparison, the iteration complexity for the exact gradient case is:
> >
> > $$
> > O\left(\frac{S \, C_{PL}^2}{c^2 (1 - \gamma)^6 \, k}\right),
> > $$
> >
> > as shown in Theorem 4 of Mei et al. (2022). We outline the high-level computation below:
>
> >> From Lemma C.1, we have:
>  $$a_k \leq c_l^{-\frac{2}{3}}\Bigm(\max\{c_\eta, 2c_l^2u^3_0\}\Bigm)^{\frac{1}{3}}\Bigm(\frac{1}{\frac{1}{\alpha^3_0}+ 2k}\Bigm)^{\frac{1}{3}}
> \leq \frac{\max\{2^{-\frac{1}{3}}c_l^{-\frac{2}{3}}c_\eta^{\frac{1}{3}}, u_0\}}{ k^{\frac{1}{3}}}.$$
>  Substituting $c_l$ and $c_\eta$ from Table 3:
>  $$c_l^{-2/3} c_\eta^{1/3} = O\left( \frac{S^{4/3} A^{2/3} C_{PL}^{4/3}}{c^{4/3} \lambda^{2/3} (1 - \gamma)^{16/3}} \right), \quad
>  u_0 = O\left( \frac{S A}{(1 - \gamma)^2} \right). $$
>  Hence, the overall complexity becomes:
>  $$
>  a_k \leq C \cdot \frac{S^{4/3} A^{2/3} C_{PL}^{4/3}}{\lambda^{2/3} (1 - \gamma)^{16/3} \, k^{1/3}},
>  $$
>  where $C$ is a numerical constant.
> Additionally, using our ODE tracking method, the iteration complexity for the exact gradient case (as in Theorem 4 of Mei et al. 2022) can be improved to:
>  $$ O\left( \frac{S \, C_{PL}^2}{c^2 (1 - \gamma)^3 \, k} \right). $$
>
> >Notably, the actor-critic complexity shows slightly *better* dependence on the mismatch coefficient $C_{PL}$ (we double-checked, decreasing step sizes improves the dependence on $C_{PL}$), and only slightly *worse* dependence on the state space size and discount horizon.
>
> > We thank the reviewer for raising this important point and will include this discussion in the final version.
>
> ***Q2) Presentation issues , typos, ... And Line 268-272 not clear.***
> > We appreciate the reviewer’s feedback on the presentation. We will address these presentation issues and revise the text to improve clarity.
>
> > Lines 268–272 highlight the challenge in ensuring a monotonic decrease in the critic error in Equation (3). Specifically, the gradient norm $y_k$ is *lower bounded* by $a_k$ due to the Gradient Domination Lemma (GDL). This lower bound on $y_k$ is useful for *upper bounding* $a_{k+1}$ in the actor recursion of Equation (3).
>
> > However, to *upper bound* the critic error $z_{k+1}^2$ in the critic recursion (also in Equation 3), we require an *upper bound* on $y_k$, which we do not have. This asymmetry presents a challenge in establishing a monotonic decrease in critic error.
>
> > We will revise this paragraph in the final version to make the reasoning clearer.
>
> ***Q3) In the discussion following Lemma 4.1, it is not clear to me why the last term “accounts for the variance in the stochastic update of the policy” since in the analysis this term arises from the smoothness of the value and not from the stochasticity of the update ?***
> > Yes, the reviewer is correct in observing that Lemma 4.1 follows from the smoothness of the return function. However, upon closer inspection of its proof in Lemma B.1, the last term is:
> >
> > $$
> > c_3 \eta_k^2 = \frac{L}{2} \, \mathbb{E}[\lVert \theta_{k+1} - \theta_k \rVert^2],
> > $$
> >
> > which corresponds to the variance (more precisely, the second moment) of the update steps.
> >
> > We thank the reviewer for noticing this subtle point. We will revise the text to state that the term “accounts for the variance (second moment) of the updates” in the final version.
>
> ***Q4) You mention that you believe that a bias analysis may lead to the optimal $O(\varepsilon^{-2})$ sample-complexity. Do you believe this is achievable without any modification to the algorithm ? Is there perhaps a slight adaptation of the algorithm that could improve your variance analysis ?***
> > This is indeed an intriguing question. Based on our recursion analysis, we believe that the current complexity is difficult to improve—our use of the ODE tracking method appears to be the most effective approach for solving the given recursion.
>
> > We also explored the use of batching, but it did not lead to an improvement in the overall complexity. However, we believe that if sample reuse were allowed, the total number of *new* samples required could potentially be reduced below the current $O(\epsilon^{-3})$ bound.
>
> > This is a promising direction, particularly from the perspective of combining offline and online reinforcement learning. We consider this an interesting avenue for future research.

---

> > ### Comment · Reviewer_ZGMW · 2025-08-02
> >
> > I thank the authors for the detailed reply. All my points have been addressed. I appreciate that improving the sample-complexity to $\varepsilon^{-3}$ remains a significant contribution.
> >
> > My main concern remains the quality of presentation, which is hard for me to assess how much that will be improved in the final version. Nevertheless, I have raised my score to 4 but I really encourage the authors to put a lot of work into improving the quality of the presentation in the next versions of this work.

---

> > > ### Author Response · Authors · 2025-08-03
> > >
> > > We sincerely thank the reviewer for the time and efforts put into reviewing the paper, and for the rasing the score. We promise to make a thorough revision the draft, in the light of reviewers feedback.

---

### Official Review · Reviewer_EEab · 2025-06-22

**Clarity:** 2
**Significance:** 4
**Originality:** 3
**Rating:** 5
**Confidence:** 4

**Summary:**

This paper studies the convergence of actor-critic algorithm, under the setting of infinite-horizon discounted MDPs. The paper considers single-timescale setting, where the actor and the critic are updated simultaneously. The paper develop a new analysis technique, which first modeling the evolution of the actor error and the critic error with a dynamic system. To bypass the difficulty incurred by the hard-to-estimate gradient norm term, the paper considers an Lyapunov term of the dynamic system. Analysis shows that the Lyapunov term is evolved according to a relative simple single-variable ODE. The solution to the ODE gives a sample complexity of $O(\epsilon^{-3})$, which strictly improves the previous $O(\epsilon^{-4})$ result.

**Questions:**

Here are my questions:

1. How does the analysis techniques compared to the analysis in previous works and why previous analysis cannot achieve the rate obtained in this paper?

2. Is there a sample-complexity lower bound for this kind of actor-critic algorithm?

**Ethical Concerns:**

["NO or VERY MINOR ethics concerns only"]

**Final Justification:**

I do not have major concerns about this paper and think that the results introduced by this paper is an important one

**Limitations:**

The author discussed the limitations in the last section.

**Quality:**

3

**Strengths And Weaknesses:**

Strengths:

1. This paper consider single-timescale actor-critic algorithm. Compared to two-timescale actor-critic, this setting is more practically relevant and hard-to-analysis.

2. The result of this paper strictly improved upon the previous results, which is new to me.

3. The technique introduced in this paper is also interesting to me. To overcome the difficulty in solving the dynamic system depicting the evolution of the errors, the paper consider the sum of actor and critic error and largely simplified the system. This technique might be inspiring to solve other similar problems.

Weakness:

I don't see any major weakness in this paper. Some part of the presentation are not clear enough. Here are some examples.

1. The 2nd and 3rd row of Table 1 looks the same.

2. In equation (3), the $z_k$ at the end of the first line seems redundant.

3. The dimension of variables is obscure. Something like $A^k \in \mathbb{R}^{|S||A|}$ might make it clearer.

4. Some references looks ill-formated. For example, it looks like that line 342-347 are referring to the same paper.

---

> ### Author Rebuttal · Authors · 2025-07-30
>
> We thank the reviewer for the positive and thoughtful feedback.
>
> We are glad that the reviewer finds our focus on the single-timescale actor-critic setting both practical and meaningful. We appreciate the recognition of our improved sample complexity and the Lyapunov-based analysis technique. We agree with the reviewer that this approach can inspire further work in similar settings.
>
>
> ***Q1) The 2nd and 3rd row of Table 1 looks the same.***
> > Thank you for pointing this out. We will merge the two in the final version.
>
> ***Q2) In equation (3), the $z_k$ at the end of the first line seems redundant.***
> > Thank you for noticing this. It is indeed a typo, and we will correct it in the final version.
>
> ***Q3) The dimension of variables is obscure. Something like $A^k \in \mathbb{R}^{|S||A|}$ might make it clearer.***
> > Thank you for the suggestion. We will include the dimensions of all variables in the final version.
>
> ***Q4) Some references looks ill-formated. For example, it looks like that line 342-347 are referring to the same paper.***
> > Thank you for pointing this out. We will fix it in the final version.
>
> ***Q5) How does the analysis techniques compared to the analysis in previous works and why previous analysis cannot achieve the rate obtained in this paper?***
>
> > The main limitation of the previous analysis by Chen and Zhao [2024] is that they treat the policy optimization objective as a general smooth non-convex function. Moreover, they follow the standard approach of bounding the *average squared norm of the gradient*. Their analysis does not leverage the gradient domination structure, or if used, it is only plugged in at the end—leading to a suboptimal convergence rate.
>
> > In contrast, the core innovation of our work is to exploit this structure directly when analyzing the iteration-wise drift of the actor.
>
> > Leveraging this additional structure required us to develop a novel methodology for solving the interdependent recursions, which in turn led to stronger results.
>
> > In summary, our analysis is more tailored to reinforcement learning, as it effectively incorporates the gradient domination structure—unlike the standard smooth optimization techniques used in the prior work.
>
> ***Q6) Is there a sample-complexity lower bound for this kind of actor-critic algorithm?***
> > The sample complexity lower bound for reinforcement learning is $O(\epsilon^{-2})$, as shown by Auer et al. (2008). However, to the best of our knowledge, there is no known lower bound specifically for single-time-scale actor-critic algorithms. Nonetheless, it is widely believed that the same lower bound applies in this setting as well.

---

> > ### Comment · Reviewer_EEab · 2025-08-01
> >
> > Thank you for your response. I will keep my score

---

### Official Review · Reviewer_FxH5 · 2025-07-02

**Clarity:** 4
**Significance:** 3
**Originality:** 4
**Rating:** 5
**Confidence:** 4

**Summary:**

This paper studies the global convergence of single-timescale actor-critic algorithms in reinforcement learning, for infinite-horizon, discounted Markov Decision Processes  with finite state and actions spaces. In a single-timescale setting, both the actor and critic are updated simultaneously using the same step-size schedule, which contrasts with the more commonly analyzed two-timescale methods that separate actor and critic updates across different learning rates or loops. This makes the analysis more challenging but brings it closer to the practically implemented. The paper achieves a sample complexity of $\mathcal{O}(\epsilon^{-3})$ which matches that of state of the art results but does so with less assumptions as compared to other works.

**Questions:**

My questions are related to the weaknesses I pointed out.
1. Is there any path to extend the analyses to infinite state/action spaces?
2. What challenges do you anticipate if any in extending your analysis to bilevel RL setup?

**Ethical Concerns:**

["NO or VERY MINOR ethics concerns only"]

**Final Justification:**

I have gone over the replies of the reviewers. They have answered the questions to my satisfaction, and I maintain my positive score.

**Limitations:**

Limitations are that the analysis is restricted to tabular state/action setup.

**Paper Formatting Concerns:**

None.

**Quality:**

3

**Strengths And Weaknesses:**

Strengths
1. The paper proves that a single-timescale AC algorithm achieves an ϵ-optimal policy with a sample complexity of $\epsilon^{-3}$. This improves upon the previous best-known  $\epsilon^{-4}$ result for such algorithms.
2. The paper achieves this under single timescale setup which makes the setup more realistic and applicable to real world applications.
3. The paper does not require the extra memory requirements for works such as Gaur et. al 2024. Making it more efficient.
4. Achieves global convergence using Lyapunov related analyses. Improving upon previous results which only got local convergence.

Weaknesses.
1. The paper is restricted to tabular setups. This means not applicable to environments such as robotics.
2. In certain situations, using multiple samples to estimate the critic can be beneficial such as DDPG. A discussion of these would be good.
3. The authors claim that the analysis may be extended to bilvel RL setup. More discussion on that would be good.

---

> ### Author Rebuttal · Authors · 2025-07-30
>
> We thank the reviewer for the encouranging feedback. We appreciate the time and effort spent on reviewing this paper. We are glad that the reviewer finds that the paper provides state-of-the-art complexity result to a challenging and more practical single-time scale algorithm.
>
> We answer the questions below.
>
>  ***Q1) The paper is restricted to tabular setups. This means not applicable to environments such as robotics.***
>  > The reviewer is correct in noting that the current setting is not directly applicable to large-state problems such as those in robotics. The main limitation lies in the square-root state dependence of Gradient Dominant Lemma (GDL) 2.1. However, we agree that extending this lemma to continuous state spaces—potentially through embedding or an approximate version—is a promising direction for future research. Such an extension would also facilitate a non-tabular generalization of the exact gradient convergence results in Xiao et al. (2022) and Mei et al. (2022). As a result, our actor-critic complexity bounds could be extended to the non-tabular setting, with minor modifications to account for function approximation errors and other approximations. We thank the reviewer for this insightful question, which we intend to pursue in future work.
>
> ***Q2) In certain situations, using multiple samples to estimate the critic can be beneficial such as DDPG. A discussion of these would be good.***
> >We thank the reviewer for this insightful question. While many existing works on double-loop actor-critic methods rely on using a large number of critic updates per actor update, our work lies at the other extreme—using a single critic update per actor update. It is quite possible that the optimal balance lies somewhere between these two extremes, either generally or in specific contexts such as DDPG. We will include a discussion of this point in the final version and explore it further as part of our future work.
>
>
> ***Q3) The authors claim that the analysis may be extended to bilvel RL setup. More discussion on that would be good.***
> >Our work proposes a high-level approach for tackling bi-level problems such as min-max optimization, robust MDPs, and two- (or multi-) agent reinforcement learning:
>
> >>**Step 1:** Formulate the interdependent recursions, as illustrated in Equation (3), for the relevant quantities.
> >>**Step 2:** Identify a suitable Lyapunov function, as demonstrated in Line 274 — often the most creative and critical part of the analysis.
> >>**Step 3:** Solve the Lyapunov recursion, as done in Lemma 4.4, to obtain the desired bound.
>
> Lemma 4.4 can be extended to handle more general recursion structures , which we chose to omit for clarity. Additionally, we uncovered several useful insights during this process — for example, that such recursions often track the behavior of the corresponding ODE, and that appropriate learning rates can be chosen using a _power-matching_ technique.
>
> We will be glad to include a dedicated section in the appendix (along with a more general ODE-tracking lemma) in the final version.
>
> ***Q4) Is there any path to extend the analyses to infinite state/action spaces?***
> > As discussed in Q1, the key step in extending the analysis to infinite state-action spaces is to establish a Gradient Domination Lemma (either exact or approximate), since this is the only component that exhibits state dependence.
>
> > The rest of the analysis—such as applying a sufficient increase lemma under function approximation—is standard in the literature. One can derive the corresponding recursion and solve it with minor modifications to the existing framework.
>
> > Loosely speaking, the $\sqrt{S}$ term in GDL Lemma 2.1 corresponds to the diameter of the policy space, i.e., $\max_{\pi, \pi' \in \Pi} \lVert \pi - \pi' \rVert_2$. In the function approximation setting, it may be possible to replace this with the diameter of the parameter space, i.e., $\max_{\theta, \theta' \in \Theta} \lVert \theta - \theta' \rVert_2$. However, this substitution requires careful investigation, which we leave for future work.
>
> ***Q5) What challenges do you anticipate if any in extending your analysis to bilevel RL setup?***
> > We briefly outlined a possible approach for bi-level problems in Q3. We believe the first step—deriving the recursions—is likely to be straightforward, though potentially laborious. Identifying a suitable Lyapunov function, however, is likely the most challenging part. Once obtained, the recursion can be solved using techniques similar to our ODE tracking lemma.
>
> > That said, finding a Lyapunov function may not always be feasible or straightforward. Fortunately, it is possible to bypass this step by directly solving multiple recursions using an extended ODE tracking lemma. This would require applying the same methodology multiple times—essentially the same arguments used to prove the Lyapunov recursion. In fact, this was our original approach before we discovered the elegant Lyapunov function presented here.
>
> > We will gladly include a detailed section on this alternative route, along with the extended ODE tracking lemma, in the appendix of the final version.

---

### Decision · Program_Chairs · 2025-09-17

**Decision:**

Accept (poster)

**Comment:**

The paper improves the state-of-the-art sample complexity bound for actor-critic algorithm in tabular MDPs from $\epsilon^{-4}$ to $\epsilon^{-3}$. Besides, the algorithm is single time-scale, more practical than many previous works that rely on double time-scale. All reviewers agree this is a significant contribution on a fundamental question.

Please improve the clarity of the paper according to the discussions with the reviewers.